# The Geometry of Refusal in Large Language Models: Concept Cones and Representational Independence

Tom Wollschläger [* 1]   Jannes Elstner [* 1]   Simon Geisler [1 2]   Vincent Cohen-Addad [3]
Stephan Günnemann [1]   Johannes Gasteiger [3 4]

## Abstract

The safety alignment of large language models (LLMs) can be circumvented through adversarially crafted inputs, yet the mechanisms by which these attacks bypass safety barriers remain poorly understood. Prior work suggests that a *single* refusal direction in the model's activation space determines whether an LLM refuses a request. In this study, we propose a novel gradient-based approach to representation engineering and use it to identify refusal directions. Contrary to prior work, we uncover multiple independent directions and even multi-dimensional *concept cones* that mediate refusal. Moreover, we show that orthogonality alone does not imply independence under intervention, motivating the notion of *representational independence* that accounts for both linear and non-linear effects. Using this framework, we identify mechanistically independent refusal directions. We show that refusal mechanisms in LLMs are governed by complex spatial structures and identify functionally independent directions, confirming that multiple distinct mechanisms drive refusal behavior. Our gradient-based approach uncovers these mechanisms and can further serve as a foundation for future work on understanding LLMs.

## 1. Introduction

The breakthrough of scaling large language models (LLMs) has led to an unprecedented leap in capabilities, driving widespread real-world adoption (OpenAI, 2022). However,

[*]Equal contribution  [1]School of Computation, Information & Technology and Munich Data Science Institute, Technical University of Munich [2]Now at Google Research [3]Google Research [4]Now at Anthropic. Correspondence to: Tom Wollschläger <tom.wollschlaeger@tum.de>.

*Proceedings of the 42nd International Conference on Machine Learning*, Vancouver, Canada. PMLR 267, 2025. Copyright 2025 by the author(s).

these advancements also introduce serious risks. As artificial intelligence becomes more powerful, it can be misused for harmful purposes, such as attacking critical infrastructure or spreading misinformation. Ensuring that these models remain aligned with human values has become a crucial research challenge (Liu et al., 2023; Schwinn et al., 2025). Despite significant progress, LLMs, like all machine learning models, remain vulnerable to adversarial attacks that can bypass alignment mechanisms and induce harmful outputs (Szegedy et al., 2014; Carlini et al., 2024).

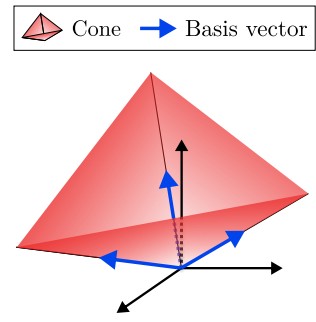

*Figure 1.* An example of a 3D concept cone with its basis vectors. All directions in the cone mediate refusal.

Recent work in interpretability has provided valuable insights into how LLMs encode and process information (Nanda et al., 2024; Wang et al., 2022; Cunningham et al., 2023; Heinzerling & Inui, 2024). Prior studies (Belrose et al., 2023; Gurnee & Tegmark, 2023; Marks & Tegmark, 2024) suggest that concepts—ranging from simple to complex—are often encoded linearly in the model's residual stream. Methods such as representation engineering (Zou et al., 2023a) allow researchers to use input prompts to analyze model behavior by extracting and manipulating such concepts. However, the mechanisms enabling adversarial jailbreaks that bypass alignment safeguards remain poorly understood. Some evidence suggests that refusals to harmful queries are mediated by a single "refusal direction" in activation space (Arditi et al., 2024), and that jailbreaks rely on manipulating this direction (Yu et al., 2024), yet these assumptions require further examination. Understanding refusal mechanisms is crucial, as it has direct implications for both offensive capabilities—informing more sophisticated jailbreaks (Huang et al., 2024)—and more importantly developing robust defensive strategies like improved adversarial training (Yu et al., 2024) and inference–time monitoring.

In this work, we go beyond extracting concepts using common input prompt methods by introducing a novel *gradient-based* approach to *representation engineering* which we use to investigate the mechanisms underlying refusal behavior in LLMs. We extract refusal-mediating directions more effectively, improving both precision and control while minimizing unintended side effects, which we demonstrate in Section 4. Unlike prior work that assumes model refusal is controlled by a single linear direction, we show in Section 5 that there exist *multi-dimensional polyhedral cones* which contain infinite refusal directions; we show an illustrative example in Figure 1. To further characterize refusal mechanisms in language models, we introduce *representational independence*, a criterion for identifying directions that remain mutually unaffected under intervention, capturing both linear and non-linear dependencies across layers. In Section 6, we demonstrate that even under this strict notion of independence, multiple complementary refusal directions exist.[1]

To summarize, our core **contributions** are:

- We show that our gradient-based representation engineering can advance general LLM understanding and specifically demonstrate its efficacy for understanding refusal mechanisms.

- We introduce representational independence, a practical framework for characterizing how different interventions interact within an LLM's activation space, and use it to find independent refusal directions.

- We show that rather than a single refusal direction, there exist multi-dimensional cones in which all directions mediate refusal.

## 2. Background

**Notation.** Let $f : \mathbb{T}^{N_{\text{seq}}} \rightarrow \Delta^{N_{\text{seq}} \times |\mathbb{T}|}$ denote a language model, where $\Delta^{|\mathbb{T}|}$ is the probability simplex over vocabulary $\mathbb{T}$. Given a prompt $p = (t_1, \ldots, t_{N_{\text{seq}}}) \in \mathbb{T}^{N_{\text{seq}}}$ consisting of tokens $t_i$, each token is first embedded: $x_i^{(0)} = \text{EMBED}(t_i)$. The model then processes the token sequence through $L$ layers, where at each layer $l = 1, \ldots, L$ and token position $i$ the following transformation is applied:

$$\tilde{\boldsymbol{x}}_i^{(l)} = \boldsymbol{x}_i^{(l)} + \text{ATTN}^{(l)}(x_{1:i}^{(l)}), \quad \boldsymbol{x}_i^{(l+1)} = \tilde{\boldsymbol{x}}_i^{(l)} + \text{MLP}(\tilde{\boldsymbol{x}}_i^{(l)})$$

The final residual stream $x_i^{(L+1)}$ is unembedded to yield logits: $\ell_i = \text{UNEMBED}(x_i^{(L+1)})$. The softmax function converts these logits into a probability distribution over tokens: $P(t \mid t_1, \ldots, t_i) = \text{softmax}(\ell_i)_t$. We omit technical details that are not critical for this work such as LayerNorm.

**Extracting refusal directions.** Paired prompts of harmful and harmless requests allow the extraction of a directional

[1]Resources & code: cs.cit.tum.de/daml/geometry-of-refusal

feature from the model's residual stream as shown by prior work (Panickssery et al., 2024; Bolukbasi et al., 2016; Burns et al., 2024). Recent studies obtain this direction by computing the *difference-in-means* (DIM) (Panickssery et al., 2024; Arditi et al., 2024; Stolfo et al., 2024) between model representations on datasets of harmful prompts $\mathcal{D}_{\text{harm}}$ and harmless prompts $\mathcal{D}_{\text{good}}$:

$$\boldsymbol{v}_i^{(l)} = \frac{1}{|\mathcal{D}_{\text{harm}}|} \left[ \sum_{p' \in \mathcal{D}_{\text{harm}}} \boldsymbol{x}_i^{(l)}(p') \right] - \frac{1}{|\mathcal{D}_{\text{safe}}|} \left[ \sum_{p \in \mathcal{D}_{\text{safe}}} \boldsymbol{x}_i^{(l)}(p) \right]$$

Here, $\boldsymbol{x}_i^{(l)}(p)$ represents the residual stream activations at position $i$, layer $l$ for input prompt $p$.

**Adversarial steering attacks.** The extracted harmfulness direction can be used to manipulate the model's refusal behavior. With white-box access, an attacker can prompt the model with harmful queries and suppress activations in the harmfulness direction, thereby reducing the model's probability of refusal. This can be done through *directional ablation* of $\boldsymbol{r}$ (where $\hat{\boldsymbol{r}}$ denotes the unit vector) (Zou et al., 2023a):

$$\tilde{\boldsymbol{x}}_i^{(l)} = \boldsymbol{x}_i^{(l)} - \hat{\boldsymbol{r}}\hat{\boldsymbol{r}}^\top \boldsymbol{x}_i^{(l)}, \tag{1}$$

which projects the residual stream to a subspace orthogonal to $\boldsymbol{r}$, or alternatively through *activation subtraction*:

$$\check{\boldsymbol{x}}_i^{(l)} = \boldsymbol{x}_i^{(l)} - \alpha \cdot \hat{\boldsymbol{r}}, \tag{2}$$

which subtracts a scaled $\boldsymbol{r}$ from the residual stream. We follow common practice to apply both operations across all token positions and ablation across all layers while doing subtraction only at a single layer (Arditi et al., 2024).

## 3. Related Work

**Adversarial attacks for LLMs.** Many studies have explored hand–crafted adversarial techniques, such as persona modulation (Shah et al., 2023), language modifications (Zhu et al., 2023), or prompt engineering using repetitions and persuasive phrasing (Rao et al., 2024). Other works take a more systematic approach, employing techniques like genetic algorithms and random search (Chen et al., 2024), discrete optimization over input tokens (Zou et al., 2023b), or gradient-based methods to identify high-impact perturbations (Geisler et al., 2024). While identifying these vulnerabilities enables adversarial fine-tuning (Xhonneux et al., 2024) or improved training through Reinforcement Learning with Human Feedback (RLHF), recent works suggest that robustness remains a challenge (Zou et al., 2023a; Schwinn et al., 2024; Geisler et al., 2024; Scholten et al., 2025).

**Interpretability of LLMs.** A parallel line of research focuses on understanding the internal mechanisms of LLMs,

as their natural language outputs provide a unique opportunity to link internal states to interpretable behaviors. Interpretability research has led to the identification of various "features"—concepts represented by distinct activation patterns (Cunningham et al., 2023)—as well as "circuits", which are subnetworks that implement a specific function or behavior. Prominent examples are backup circuits (Nanda et al., 2024) and information mover circuits (Wang et al., 2022). Many interpretability insights rely on extracting features using paired inputs with opposing semantics (Burns et al., 2024) and then manipulating residual stream activations to elicit specific behaviors (Panickssery et al., 2024). Representation engineering, as proposed by Zou et al. (2023a), investigates the linear representation of concepts such as truthfulness, honesty, and fairness in LLMs. The effectiveness of these methods supports the hypothesis that many features are encoded linearly in LLMs (Marks & Tegmark, 2024). These insights allow researchers to pinpoint and manipulate concept representations or specific circuits, enabling targeted debugging of behaviors, mitigating biases, and advancing safer, more reliable AI systems.

**Understanding Refusal Mechanisms.** Recent research has focused on understanding the mechanisms underlying refusal behaviors in LLMs. For example, removing safety-critical neurons has been shown to decrease robustness (Wei et al., 2024; Li et al., 2024b). Zheng et al. (2024) demonstrate that adding explicit safety prompts shifts the internal representation along a harmfulness direction. O'Brien et al. (2024) propose to use sparse autoencoders to identify latent features that mediate refusal. The most relevant work to ours is Arditi et al. (2024), which builds on Zou et al. (2023a) and examines the representation of refusal in LLMs. Their work suggests that a single direction in a model's activation space determines whether the model accepts or refuses a request. We challenge this notion by showing that refusal is mediated through more nuanced mechanisms. Concurrently, Pan et al. (2025) identify multiple independent refusal directions, providing more evidence to our findings in Section 6. While their focus is on representation shifts during safety finetuning, our work introduces a novel, gradient-based, top-down discovery approach applicable to any model.

**Over-refusal** Recent work addresses LLM over-refusal, where benign queries are rejected by overly strict safety filters. Röttger et al. (2023) (XSTest) and An et al. (2024) (PHTest) provide benchmarks to quantify false refusal rates, expanded by Cui et al. (2025)'s OR-Bench. Shi et al. (2024) attribute this to lexical shortcuts, proposing self-contrastive decoding for mitigation. Similarly, Li & Liu (2025) highlight over-defense in prompt injection guards with NotInject and InjecGuard. Collectively, these studies underscore the challenge of balancing safety and helpfulness, offering resources to diagnose and mitigate overrefusal.

## 4. Gradient–based Refusal Directions

> Research Question: Can our gradient–based representation engineering identify refusal directions?

To investigate the refusal mechanisms in language models, we propose a gradient-based algorithm that identifies directions controlling refusal in the model's activation space. We refer to it as Refusal Direction Optimization *(RDO)*. Unlike prior approaches that extract refusal directions using paired prompts of harmless and harmful instructions (Arditi et al., 2024), our method leverages gradients to find better directions instead of solely relying on model activations. Similar to (Park et al., 2023), we define two key properties for refusal directions:

**Definition 4.1.** Refusal Properties:

- *Monotonic Scaling:* when using the direction for activation addition/subtraction $\check{x}_i^{(l)} = x_i^{(l)} + \alpha \cdot r$, the model's probability of refusing instructions should scale monotonically with $\alpha$.

- *Surgical Ablation:* ablating the refusal direction through projection $\tilde{x}_i^{(l)} = x_i^{(l)} - \hat{r}\hat{r}^\top x_i^{(l)}$ should cause the model to answer previously refused harmful prompts, while preserving normal behavior on harmless inputs.

We can encode the desired refusal properties into loss functions, allowing us to find corresponding refusal vectors $r$ using gradient descent. For the monotonic scaling property, we train the model to refuse harmless instructions $p_{\text{safe}}$ when running the model $f$ with a modified forward pass $f_{\text{add}(r,l)}$ in which we add $r$ to the activations at layer $l$. We minimize the cross-entropy between the model output and target refusal response $t_{\text{refusal}}$. For the surgical ablation property, we similarly compute the cross–entropy between a harmful response target $t_{\text{answer}}$ and the output of a modified forward pass $f_{\text{ablate}(r)}$ to make the model respond to harmful instructions. A key strength of our gradient–based approach is the ability to control any predefined objective and thus we can control the extent to which other concepts are affected during interventions. For this, we use a retain loss based on the Kullback–Leibler (KL) divergence to ensure that directional ablation of $r$ on harmless instructions does not change the model's output over a target response $t_{\text{retain}}$. Algorithm 1 shows the full training procedure for our refusal directions.

**Setup.** We construct a dataset of harmless and harmful prompts from the ALPACA (Taori et al., 2023) and SALAD-BENCH (Li et al., 2024a) datasets (see Appendix A.1). An important consideration for our algorithm is the choice of targets $t_{\text{answer}}$ and $t_{\text{refusal}}$. Generally, language models differ in their refusal and response styles, which is why we use

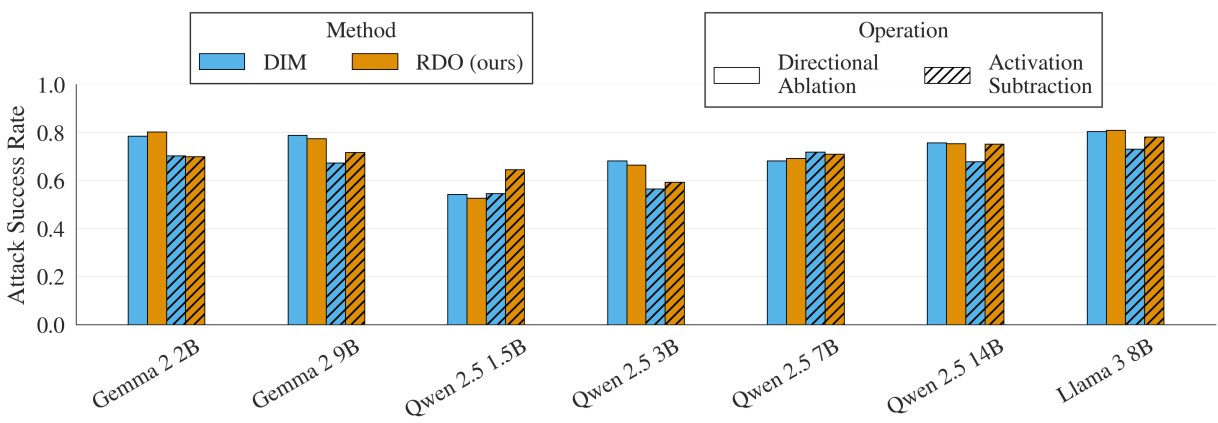

*Figure 2.* Attack success rates of refusal directions for different models. We compare the DIM direction baseline that is extracted from prompts against our Refusal Direction Optimization for jailbreaking with directional ablation and activation subtraction.

---

**Algorithm 1** Refusal Direction Optimization (RDO)

**Input:** Frozen model $f$, scaling coefficient $\alpha$, addition layer index $l_{add}$, learning rate $\eta$, loss weights $\lambda_{abl}$, $\lambda_{add}$, $\lambda_{ret}$, and data $D = \{(p_{harm,i}, p_{safe,i}, t_{answer,i}, t_{refusal,i}, t_{retain,i})\}_{i=1}^{N}$.
**Output:** Refusal direction $r$

1: **Initialize** $r$ randomly
2: **while** not converged **do**
3:    Sample batch $B \sim \mathcal{D}$
4:    $\mathcal{L} \leftarrow \text{COMPUTELOSS}(r, f, B)$
5:    $r \leftarrow r - \eta \nabla_r \mathcal{L}$
6:    $r \leftarrow r / \|r\|_2$
7: **end while**

1: **function** COMPUTELOSS($r, f, B$)
2:    $p_{harm}, p_{safe}, t_{answer}, t_{refusal}, t_{retain} = B$
3:    $\mathcal{L}_{ablation} = \text{CELOSS}(f_{ablate(r)}(p_{harm}), t_{answer})$
4:    $\mathcal{L}_{addition} = \text{CELOSS}(f_{add(\alpha\hat{r}, l_{add})}(p_{safe}), t_{refusal})$
5:    $\mathcal{L}_{retain} = \text{KL}(f_{ablate(r)}(p_{safe}), f(p_{safe}), t_{retain})$
6:    $\mathcal{L} = \lambda_{abl}\mathcal{L}_{ablation} + \lambda_{add}\mathcal{L}_{addition} + \lambda_{ret}\mathcal{L}_{retain}$
7:    **return** $\mathcal{L}$

---

model–specific targets rather than generating them via uncensored LLMs as in Zou et al. (2024). Specifically, we use the DIM refusal direction to generate our targets, though any effective attack can work. For the harmful answers $t_{answer}$, we ablate the DIM direction and generate 30 tokens. Similarly, we use activation addition on harmless instructions to produce refusal targets $t_{refusal}$. For helpful answers on harmless instructions that should be retained $t_{retain}$, we generate 29 tokens without intervention. The retain loss $\mathcal{L}_{retain}$ is applied over the last 30 tokens, such that the last token of the model's chat template is included. We detail hyperparameters and implementation in Appendix A.

**Evaluation.** We evaluate our method by training a refusal direction on various models from the Gemma 2 (Team et al.,

2024), Qwen2.5 (Yang et al., 2024), and Llama-3 (Dubey et al., 2024) families and compare against the DIM direction for which we use the same setup as Arditi et al. (2024) but with our expanded dataset. For a fair comparison, we train the refusal direction at the same layer that the DIM direction is extracted from, and during activation addition/subtraction set the scaling coefficient $\alpha$ to the norm of the DIM direction. We evaluate the jailbreak Attack Success Rate (ASR) on JAILBREAKBENCH (Chao et al., 2024) using the STRONGREJECT fine–tuned judge (Souly et al., 2024). For inducing refusal via activation addition, we test 128 harmless instructions sampled from ALPACA using substring matching of common refusal phrases. Model completions for evaluation are generated using greedy decoding with a maximum generation length of 512 tokens.

**Does the direction mediate refusal?** In Figure 2, we show that for jailbreaking, our approach is competitive when using directional ablation and, on average, outperforms DIM when subtracting the refusal direction. Notably, despite not being explicitly optimized for subtraction–based attacks, our direction naturally generalizes to this setting. Figure 10 shows that adding the refusal direction to harmless inputs induces refusal more effectively with RDO than with DIM, further indicating that our method manipulates refusal more effectively.

**Is the direction more precise?** To measure the side effects when intervening with the directions we track benchmark performance. Arditi et al. (2024) show that directional ablation with the DIM direction tends to have little impact on benchmark performance, except for TruthfulQA (Lin et al., 2021). In Table 1, we show that RDO impacts TruthfulQA performance much less severely, reducing the error by 40% on average. We show the results for more benchmarks in Appendix B.3.

We then evaluate the trade–off between safety and over–

*Table 1.* TruthfulQA benchmark performance for directional abla-
tion with the DIM or RDO directions, compared to the baseline
(no intervention). Higher values indicate better performance.

| Chat model | DIM | RDO (ours) | Baseline |
|---|---|---|---|
| GEMMA 2 2B | 47.8 | 51.4 (+3.6) | 55.8 |
| GEMMA 2 9B | 52.8 | 56.7 (+3.9) | 61.1 |
| LLAMA 3 8B | 48.7 | 51.0 (+2.3) | 52.8 |
| QWEN 2.5 1.5B | 42.9 | 44.0 (+1.1) | 46.5 |
| QWEN 2.5 3B | 54.2 | 54.5 (+0.3) | 57.2 |
| QWEN 2.5 7B | 58.7 | 60.0 (+1.3) | 63.1 |
| QWEN 2.5 14B | 63.3 | 67.9 (+4.6) | 70.8 |

refusal for RDO and DIM on the XSTest benchmark
(Röttger et al., 2023). As detailed in Appendix G, RDO
consistently achieves a higher refusal rate for harmful in-
puts while maintaining or reducing the benign over–refusal
rate compared to DIM. This means that for any given level
of benign over–refusal, our method refuses more harmful
requests, thereby yielding a uniformly better trade–off.

**Is our method versatile?** Hyperparameter tuning of the
retain loss weight $\lambda_{ret}$ in Algorithm 1 allows for balancing
between attack success and side effects (Appendix C.2). We
observe that for many models—especially those in the Qwen
2.5 family—the majority of estimated DIM directions have
too high side–effects, rendering it an unsuccessful attack
(Figure 16). Our method is more flexible than previous work
as we can choose the target layer freely while limiting side
effects through the retain loss.

> **Key Takeaways.** Our RDO yields more effective re-
> fusal directions with fewer side effects, establishing
> that gradient–based representation engineering is an
> effective approach for extracting meaningful direc-
> tions, while allowing for more modeling freedom
> such as incorporating side constraints.

## 5. Multi-dimensional Refusal Cones

> Research Question: Is refusal in LLMs governed by
> a single direction, or does it emerge from a more
> complex underlying geometry?

We extend RDO to higher dimensions by searching for
regions in activation space where all vectors control re-
fusal behavior. For this, we optimize an orthonormal basis
$\mathcal{B} = [\boldsymbol{b}_1, \ldots, \boldsymbol{b}_N]$ spanning an $N$-dimensional polyhedral
cone $\mathcal{R}_N = \{\sum_{i=1}^{N} \lambda_i \boldsymbol{b}_i \mid \lambda_i \geq 0\} \backslash \{\boldsymbol{0}\}$, where all direc-
tions $\boldsymbol{r} \in \mathcal{R}_N$ satisfy the refusal properties (Definition 4.1).
Since all directions in the cone correspond to the same re-
fusal concept, we also refer to this as a *concept cone*. The

constraint $\lambda_i \geq 0$ ensures that all directions within the
cone consistently strengthen refusal behavior. Without this
constraint, allowing negative coefficients could introduce
opposing effects, reducing the overall effectiveness. En-
forcing orthogonality of the basis vectors prevents finding
co-linear directions. Note that in practice, directions in acti-
vation space cannot be scaled arbitrarily high without model
degeneration, which effectively bounds $\lambda_i$.

---

**Algorithm 2** Refusal Cone Optimization (RCO)

1: **Initialize** $\mathcal{B} = [\boldsymbol{b}_1, \ldots, \boldsymbol{b}_n]$ randomly
2: **while** not converged **do**
3:      Sample batch B $\sim \mathcal{D}$
4:      $\mathcal{L}_{sample} \leftarrow \mathbb{E}_{\boldsymbol{r} \sim Sample(\mathcal{B})}[\text{COMPUTELOSS}(\boldsymbol{r}, f, \text{B})]$
5:      $\mathcal{L}_{basis} \leftarrow \frac{1}{n} \sum_{i=1}^{n} \text{COMPUTELOSS}(\boldsymbol{b}_i, f, \text{B})$
6:      $\mathcal{L} = \mathcal{L}_{sample} + \mathcal{L}_{basis}$
7:      $\mathcal{B} \leftarrow \mathcal{B} - \eta \nabla_{\mathcal{B}} \mathcal{L}$
8:      $\mathcal{B} \leftarrow \text{GRAMSCHMIDT}(\mathcal{B})$
9: **end while**

1: **function** SAMPLE($\mathcal{B}$)
2:      $\boldsymbol{s} \sim \text{Unif}(\boldsymbol{x} \in \mathbb{R}_+^n : ||\boldsymbol{x}||_2 = 1)$
3:      $\boldsymbol{r} = \mathcal{B}\boldsymbol{s}$
4:      **return** $\boldsymbol{r}$

---

In Algorithm 2, we describe the procedure to find the cone's
basis vectors. The basis vectors are initialized randomly
and iteratively optimized using projected gradient descent.
We compute the previous losses defined in Algorithm 1 on
Monte Carlo samples from the cone, as well as on the ba-
sis vectors themselves. Computing the loss on the basis
vectors improves both stability and the lower bounds of
the ASR. This is because the basis vectors are the bound-
aries of the cone and thus tend to degrade first. After each
step, we project the basis back onto the cone using the
Gram–Schmidt orthogonalization procedure. Because the
directional ablation operation uses the normalized $\hat{\boldsymbol{r}}$ rather
than $\boldsymbol{r}$, sampling convex combinations of the basis vectors
and normalizing them would introduce a bias towards the
basis vectors themselves. Instead, we sample unit vectors in
the cone uniformly to ensure better coverage of the space.

**Can we find refusal concept cones?** We train cones of in-
creasing dimensionality using the same experimental setup
as described in Section 4. We measure the cone's effective-
ness in mediating refusal by sampling 256 vectors from each
cone and computing the ASRs of the samples for directional
ablation. We show the results in Figure 3 and confirm that
the directions in the cones have the desired refusal proper-
ties in Figure 20. Notably, we identify refusal–mediating
cones with dimensions up to five across all tested models.
This suggests that the activation space in language models
exhibits a general property where refusal behavior is en-
coded within multi–dimensional cones rather than a single

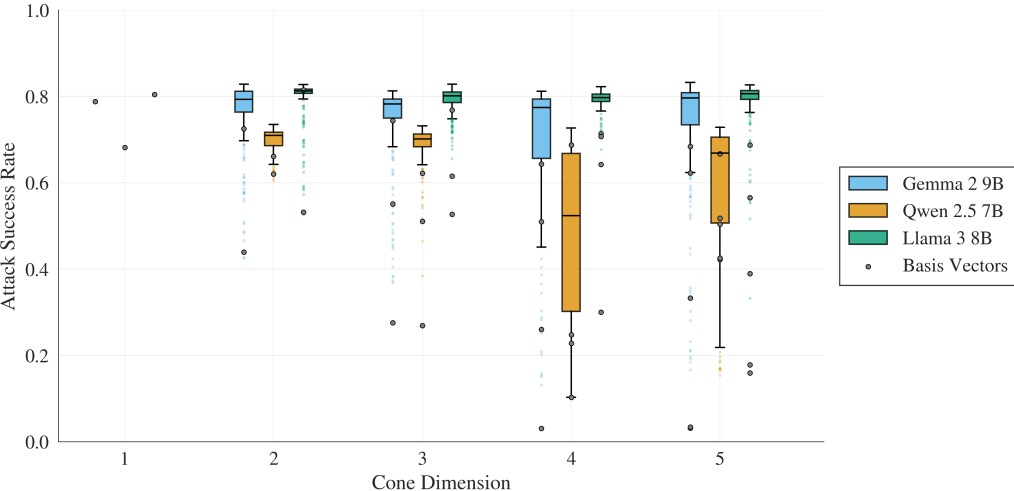

*Figure 3.* Attack success rate for multi-dimensional cones for Gemma 2, Qwen 2.5 and Llama 3. The cone performance is measured via the performance of Monte Carlo samples which are depicted as boxplot.

linear direction.

**Do larger models contain higher–dimensional cones?**
In Figure 4, we evaluate the effect of model size within the Qwen 2.5 family. We observe that across all model sizes, the lower bounds of cone performance degrade significantly as dimensionality increases. In other words, a higher number of sampled directions have low ASR. Larger models appear to support higher–dimensional refusal cones. A plausible explanation is that models with larger residual stream dimensions (e.g., 5120 for the 14B model vs. 1536 for the 1.5B model) allow for more distinct and orthogonal directions that mediate refusal. Finally, in Figure 18, we confirm that directions sampled from these cones effectively induce refusal behavior, further supporting the notion that multiple axes contribute to the model's refusal decision.

**Do different directions uniquely influence refusal?**
To further investigate the role of different vectors, we assess whether multiple sampled cone directions influence the model in complementary ways. Specifically, we sample varying numbers of directions from Gemma–2–2B's four–dimensional refusal cone and, for each prompt, select the most effective one under directional ablation (more details in Appendix A). To ensure a fair comparison, we use temperature sampling with the single–dimension RDO direction to generate the same number of attacks and similarly select the most effective instance. We study Gemma 2 2B and sample from its four–dimensional cone, since performance degrades significantly for larger dimensions (see Figure 17).

Figure 5 shows that sampling multiple directions leads to higher ASR compared to sampling with various temperatures in the low–sample regime. For a higher number of samples, the randomness dominates the success of the at-

tack. However, the higher ASR in the low–sample regime suggests that different directions capture distinct, complementary aspects of the refusal mechanism. Additionally, Figure 19 reveals that ASR increases with cone dimensionality but plateaus at four dimensions. This trend indicates that higher–dimensional cones offer an advantage over single–direction manipulation, likely by influencing complementary mechanisms. The plateau likely occurs because the model does not support higher–dimensional refusal cones.

> **Key Takeaways.** We show that refusal mechanisms in LLMs span high–dimensional polyhedral cones, capturing diverse aspects of refusal behavior. This highlights their geometric complexity and demonstrates the effectiveness of our gradient–based method in identifying intricate structures.

## 6. Mechanistic Understanding of Directions

> Research Question: Are there genuinely independent directions that influence a model's refusal behavior? Can we access the discovered refusal directions through perturbations in the token space?

In the previous section, we demonstrated that refusal behavior spans a multi–dimensional cone with infinitely many directions. However, whether the orthogonal refusal–mediating basis vectors manipulate independent mechanisms remains an open question. In this section, we conduct a mechanistic analysis to investigate how these directions interact within the model's activation space and whether they can be directly influenced through input manipulation.

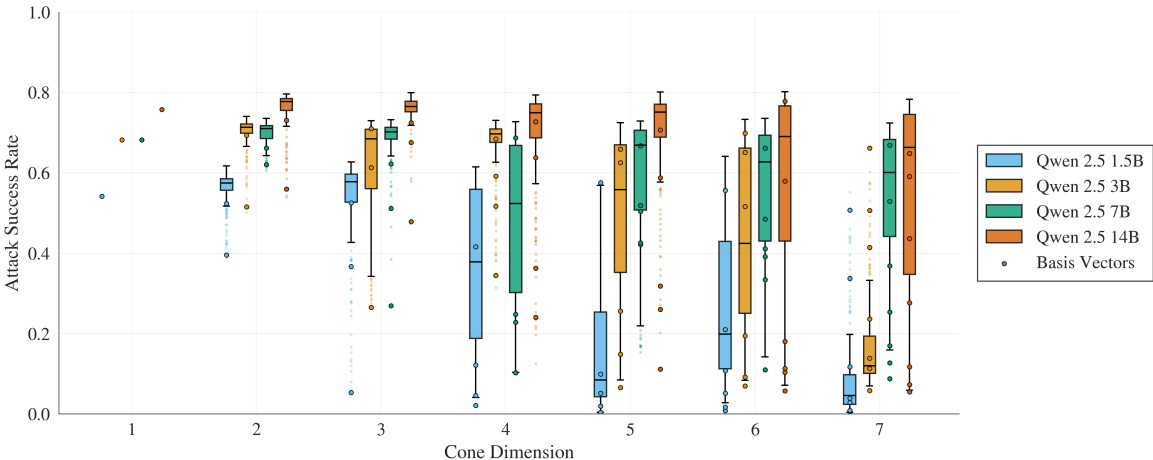

*Figure 4.* Refusal evaluation for different cone dimensions for the Qwen2.5 model family. The cone performance for models with fewer parameters degrades faster with increasing cone dimension compared to larger models.

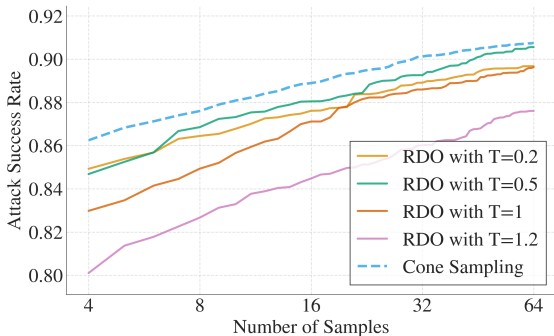

*Figure 5.* ASR for best–of–N sampling using $N$ samples from the 4–dimensional refusal cone of Gemma–2–2B, compared to best–of–N sampling with temperature $T$ using the single–dimension RDO.

This allows us to determine whether they are merely latent properties of the network or actively utilized by the model in response to specific prompts.

### 6.1. Representational Independence

We defined the basis vectors of the cones to be orthogonal, which is often considered an indicator of causal independence. The intuition is that if two vectors are orthogonal, they each influence a third vector without interfering with the other. Mathematically, for the directions $r$, $v$ and representation $x_i^{(l)}$ we have:

$$\text{if } r^T v = 0 \text{ then } r^T(x_i^{(l)} - vv^T x_i^{(l)}) = r^T x_i^{(l)}.$$

However, despite this mathematical property, recent work by Park et al. (2024) suggests that in language models, conclusions about causal independence cannot be drawn using orthogonality measured with the Euclidean scalar product. Al-

though their assumptions differ from ours, especially since they assume a one–to–one mapping from output feature to direction in activation space, their experiments suggest that independent directions are almost orthogonal. This motivates a deeper empirical examination of how orthogonal refusal directions in language models interact in practice.

**Are orthogonal directions independent?** To explore this, we first use RDO to identify a direction $r$ that is orthogonal to the DIM direction $v$, i.e., $r^\top v = 0$. We then measure how much one direction is influenced when ablating the other direction by monitoring the cosine similarity $\cos(\lambda, \mu) = \frac{\lambda^\top \mu}{||\lambda|| \cdot ||\mu||}$ between the prompt's representation in the residual stream $x$ and the directions $v$ and $r$. Specifically, we track: $\cos(r, x_i^{(l)}(p_{\text{harm}}))$ and $\cos(v, x_i^{(l)}(p_{\text{harm}}))$ at the last token position and for all layers $l \in \{0, \ldots, L\}$ on 128 harmful instructions in our validation set. Intuitively, ablating a causally independent direction in earlier layers should not intervene with the reference direction in later layers. Otherwise, there is some indirect influence through the non–linear transformations of the neural network.

The top row of Figure 6 shows how the cosine similarity between the RDO and DIM directions changes under intervention. The left plot shows the cosine similarity between the RDO direction and the activations on a normal forward pass (solid line) and while ablating the DIM direction (dashed line). The right plot presents the reverse setting. Despite enforced orthogonality, ablating RDO indirectly reduces the representation of the DIM direction in the model activations in the later layers, as measured by cosine similarity. This effect is reciprocal, suggesting that orthogonality alone does not guarantee independence throughout the network. In Figure 21 we show the results for additional models.

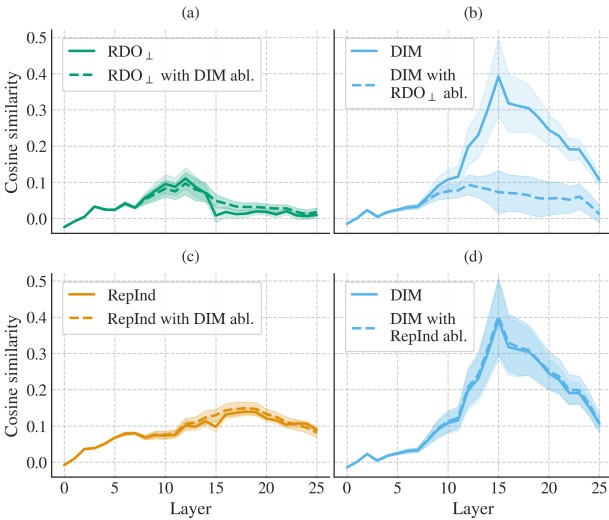

Figure 6. Influence of representational independence. Figure (a) shows the cosine similarity between RDO$_\perp$, a refusal direction orthogonal to DIM, and the model activations in a normal forward pass (solid line) compared to a forward pass where DIM is ablated (striped line). Figure (b) shows the reverse scenario. In Figure (c) and (d) we contrast how the DIM direction and a representationally independent direction (RepInd) influence each other.

Motivated by this observation, we introduce a stricter notion of independence: *Representational Independence (RepInd)*:

**Definition 6.1.** The directions $\lambda, \mu \in \mathbb{R}^d$ are *representationally independent* (under directional ablation) with respect to the activations $\boldsymbol{x}$ of a model in a set of layers $l \in L$ if:

$$\forall l \in L : \cos(\boldsymbol{x}^{(l)}, \lambda) = \cos(\hat{\tilde{\boldsymbol{x}}}^{(l)}_{abl(\mu)}, \lambda)$$
$$\text{and } \cos(\boldsymbol{x}^{(l)}, \mu) = \cos(\hat{\tilde{\boldsymbol{x}}}^{(l)}_{abl(\lambda)}, \mu),$$

where $\hat{\tilde{\boldsymbol{x}}}^{(l)}_{abl(\lambda)} = \left( f^{(l)}(\hat{\tilde{\boldsymbol{x}}}^{(l-1)}_{abl(\lambda)}) + \hat{\tilde{\boldsymbol{x}}}^{(l-1)}_{abl(\lambda)} \right)_{abl(\lambda)}$ denotes the activations at layer $l$ produced from the previous layer's *already ablated* activations with the ablation applied again after the residual addition.

Instead of relying solely on geometric orthogonality, we say that two directions are representationally independent when ablating one of them does not change how strongly the other is expressed in the model's activations. Because we track cosine similarity at every layer, any non–linear distortions introduced earlier in the network are captured downstream. Consequently, representational independence guarantees that—measured by cosine similarity—no linear, non–linear, or cumulative interaction in the network increases or decreases how much the other examined direction is represented.

To enforce this property, we extend Algorithm 1 with an additional loss term that penalizes changes in cosine similarity at the last token position when ablating on harmful

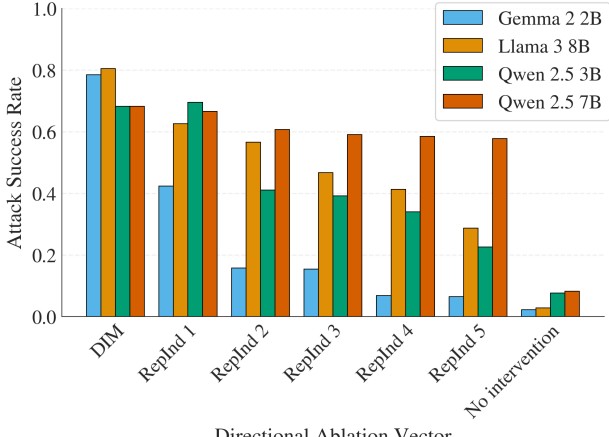

Figure 7. Attack success rate for jailbreaking the model with directional ablation of representationally independent refusal directions for different models on JAILBREAKBENCH. Each direction is representationally independent to all previous directions and the DIM direction.

instructions:

$$\mathcal{L}_{\text{RepInd}} = \frac{1}{|L|} \sum_{l \in L} \left[ \left( \cos(\boldsymbol{x}^{(l)}, \boldsymbol{r}) - \cos(\hat{\tilde{\boldsymbol{x}}}^{(l)}_{abl(\boldsymbol{v})}, \boldsymbol{r}) \right)^2 \right.$$
$$\left. + \left( \cos(\boldsymbol{x}^{(l)}, \boldsymbol{v}) - \cos(\hat{\tilde{\boldsymbol{x}}}^{(l)}_{abl(\boldsymbol{r})}, \boldsymbol{v}) \right)^2 \right].$$

**Do independent directions exist?** With this extension, we can find a direction that is RepInd from the DIM direction, yet still fulfills the refusal properties from Definition 4.1. We illustrate the representational independence for Gemma 2 2B in the second row of Figure 6, where we see that the RepInd and DIM direction barely affect each other's representation under directional ablation.

We iteratively search for additional directions that are not only RepInd to DIM but also of all previously identified RepInd directions. Despite these strong constraints, we successfully identify multiple such directions that maintain an ASR significantly above random vector intervention (Figure 7). The ASR declines as you search for more directions, which could be attributed to the increased difficulty of the optimization problem due to additional constraints, or that the models contain a limited number of directions that independently contribute to refusal. Nevertheless, these results show that refusal in LLMs is mediated by multiple *independent* mechanisms, underpinning the idea that refusal behavior is more nuanced than previously assumed.

**Do the directions manipulate different mechanisms?** Representational independence should have causal significance in language models, such that ablating different representationally independent directions corresponds to manipulating independent mechanisms. For this, we demonstrate that simultaneously ablating multiple representationally independent directions yields better performance. Figure 8

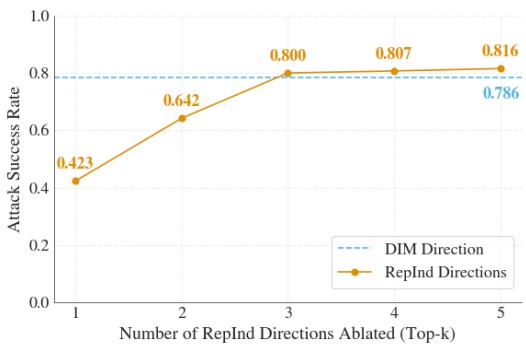

*Figure 8.* Compositionally ablating the top–k RepInd directions compared to ablating the DIM direction

shows that when ablating the top–k RepInd directions for Gemma 2 2B, the attack success rate increases monotonically with k, even surpassing the DIM baseline for k≥4, though with diminishing returns beyond this point. In contrast, we found that ablating multiple DIM directions extracted from different layers does not improve performance. The fact that ablating multiple RepInd directions produces additive improvements in ASR provides evidence that they capture different aspects of refusal rather than different manifestations of a single mechanism.

### 6.2. Manipulation from input

**Can we access these directions from the input?** Having found several independent directions that are distinct from DIM, we investigate whether these directions can ever be "used" by the model, by checking if they are accessible from the input or if they live in regions that no combination of input tokens activates. To this end, we use GCG (Zou et al., 2023b) to train adversarial suffixes, which are extensions to the prompts that aim to circumvent the safety alignment. In addition to the standard cross–entropy loss on an affirmative target, we add a loss term that incentivizes the suffix to ablate RepInd–1.

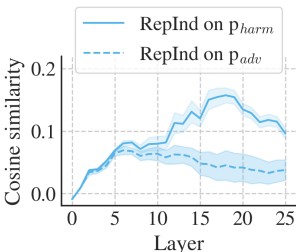

In Figure 9, we show the cosine similarities between RepInd–1 and the model activations on both harmful prompts $p_{harm}$ from JAILBREAK-BENCH and the same prompts with adversarial suffixes $p_{adv}$. We observe that GCG is able to create suffixes that significantly reduce how much RepInd–1 is represented. These suffixes successfully jailbreak the model 36% of the time, which is similar to the ASR of RepInd–1.

*Figure 9.* Representation of the RepInd–1 direction in model activations on harmful instructions before and after adversarial attacks with GCG.

> **Key Takeaways.** We demonstrate the ability to identify independent refusal directions, revealing that these directions correspond to distinct underlying concepts and can be directly accessed through input manipulations. This further underscores the utility of our representational independence framework, which provides a generalizable approach for analyzing and understanding a wide range of representational interventions in LLMs.

## 7. Limitations

While our work provides new insights into the geometry of refusal in LLMs, some limitations remain. The refusal directions we compute are all optimized on the same targets, which may limit their ability to capture fully distinct mechanisms. Extending our method to incorporate diverse targets or leveraging reinforcement learning with a judge-based reward function could help identify additional independent mechanisms (Geisler et al., 2025). Furthermore, while we establish the existence of higher-dimensional refusal cones, we cannot rule out the possibility of other yet-undiscovered regions in the model that mediate refusal.

## 8. Conclusion

This work advances the understanding of refusal mechanisms in LLMs by introducing gradient-based representation engineering as a powerful tool for identifying and analyzing refusal directions. Our method yields more effective refusal directions with fewer side effects, demonstrating its viability for extracting meaningful structures while allowing for greater modeling flexibility. We establish that refusal behaviors can be better understood via high-dimensional polyhedral cones in activation space rather than a single linear direction, highlighting their complex spatial structures. Additionally, we introduce representational independence and show that within this space of independent directions multiple refusal directions exist and correspond to distinct mechanisms. Our gradient-based representation engineering approach can be extended to identify various concepts beyond refusal simply by changing the optimization targets. The generated findings provide new insights into the geometry of aligned LLMs, highlighting the importance of structured, gradient-based approaches in LLM interpretability and safety.

## Acknowledgements

This project was conducted in collaboration with and supported by funding from Google Research. We thank Dominik Fuchsgruber and Leo Schwinn for feedback on an early version of the manuscript.

## Impact Statement

Understanding how refusal mechanisms in language models work could potentially aid adversaries in developing more effective attacks. However, our research aims to deepen the understanding of refusal mechanisms to help the community develop more robust and reliable safety systems. By focusing on open-source models requiring white-box access, our findings are primarily applicable to improving defensive capabilities rather than compromising deployed systems. We believe the positive impact of advancing model alignment and safety through better theoretical understanding outweighs the potential risks, making this research valuable to share with the research community.

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

# A. Setup Details

## A.1. Datasets

We construct our experimental dataset using harmful and harmless instructions from established benchmarks. For harmful instructions, we draw from SALADBENCH (Li et al., 2024a), a comprehensive collection of adversarial prompts from diverse sources. We exclude the Multilingual (Wang et al., 2023) and ToxicChat (Lin et al., 2023) sources since they are unsuited as harmful instructions. Afterwards, we sample up to 256 instructions from each remaining source. This results in 1,184 instructions for training and 128 for validation. We sample equal numbers of harmless instructions from the ALPACA dataset, and additionally reserve 128 more harmless instructions for testing.

## A.2. Models

We exclusively use chat models for our experiments, but omit "IT" and "INSTRUCT" from model names. We use each chat model's default chat template throughout our analysis.

*Table 2.* Model families, sizes, and references.

| Model family | Sizes | Reference |
|---|---|---|
| QWEN2.5 INSTRUCT | 1.5B, 3B, 7B, 14B | Yang et al. (2024) |
| GEMMA 2 IT | 2B, 9B | Team et al. (2024) |
| LLAMA-3 INSTRUCT | 8B | Dubey et al. (2024) |

## A.3. Hyperparameters and Implementation

*Table 3.* Hyperparameters for all algorithms

| Component | Parameter | Value |
|---|---|---|
| Training | Total Batch Size | 16 |
| | Gradient Accumulation Steps | 16 |
| | Base Learning Rate | 0.01 |
| | Learning Rate Reduction | Every 5 batches if plateaued |
| | Learning Rate Factor | Divide by 1/10 up to 2 times |
| | Optimizer | AdamW |
| | Weight Decay | 0 |
| Main Loss | Ablation Loss Weight $\lambda_{abl}$ | 1.0 |
| | Addition Loss Weight $\lambda_{add}$ | 0.2 |
| | Retain Loss Weight $\lambda_{ret}$ | 1.0 |
| Monte Carlo Sampling | Samples per Accumulation Step | 16 |
| | Effective Samples per Batch | 256 |
| RepInd | RepInd Loss Weight $\lambda_{ind}$ | 200 |
| | Layer Cutoff | 0.9 |

Table 3 presents the hyperparameters used in our algorithms. Since our method converges before completing a full epoch, we do not utilize validation scores during training. Instead, after convergence, we apply the direction selection algorithm from Arditi et al. (2024) to identify the optimal refusal direction from the last 20 training steps.

**Implementation and Evaluation Framework.** All algorithms and exploratory experiments are implemented using the NNsight (Fiotto-Kaufman et al., 2024) library. Additionally, we use the LM Evaluation Harness (Gao et al., 2024) to run TruthfulQA (Lin et al., 2021) with default settings, with the exception that we enable the use of each model's default chat templates.

**Retain and Representational Independence Loss Computation.** The retain loss is computed as the KL divergence between the probability distributions derived from the logits of the model with and without directional ablation, masked

over a target response and the last token of the chat template. The resulting value is then averaged across tokens. For a single instruction $p_{\text{safe}}$ with its target $t_{\text{retain}}$, we formalize the loss as follows:

$$\mathcal{L}_{\text{retain}} = \text{KL}(f_{\text{ablate}(\boldsymbol{r})}(p_{\text{safe}}), f(p_{\text{safe}}), t_{\text{retain}}) = \frac{1}{|\mathcal{I}|} \sum_{i \in \mathcal{I}} \sum_{t \in \mathbb{T}} f(p_{\text{safe}} + t_{\text{retain}})_{i,t} \log \frac{f(p_{\text{safe}} + t_{\text{retain}})_{i,t}}{f_{\text{ablate}}(p_{\text{safe}} + t_{\text{retain}})_{i,t}} ,$$

where $\mathcal{I}$ contains the target token indexes and the last instruction token's index, the subscript $i, t$ denotes the model output at sequence position $i$ and vocabulary index $t$ as defined in Section 2, and $+$ denotes concatenation.

For the implementation of the representational independence loss, $\mathcal{L}_{\text{RepInd}}$, we compute the average loss over the tokens in the harmful instructions $p_{\text{harm}}$. The RepInd loss is computed over the first 90% of layers, as applying it too close to the unembedding layer overly constrains the model's output.

**Selection of Refusal and Independent Directions** In Algorithm 1, after training the refusal directions to convergence, we again use the direction selection algorithm from Arditi et al. (2024) to identify the most effective directions from the final 20 training steps.

In Section 5, we extend this selection process to determine a basis where all basis vectors effectively mediate refusal (from the last 20 bases of the training). If no such basis exists, we instead select the basis where the samples are most effective for directional ablation using the refusal score heuristic from the selection algorithm.

**Training Procedure for Representational Independence Directions** In Section 6, our approach to training and validating representationally independent (RepInd) directions differs because of high variance between different runs. For each RepInd direction, we train five candidate vectors and select the one with the lowest refusal score on our validation set. This process is repeated five times, ultimately producing our final set of RepInd directions. The RepInd loss is computed as the sum of losses over all vectors that the current vector should remain independent of.

# B. Extended Results for Refusal Direction Optimization

In this section, we present additional results, including results for activation addition, more datasets for directional ablation, and more benchmarks.

## B.1. Activation Addition

We first confirm that our directions can also be used to induce refusal. Figure 10 demonstrates that using RDO refusal directions for activation addition successfully induces refusal behavior across all models for both DIM and RDO, and RDO slightly outperforms DIM for most models.

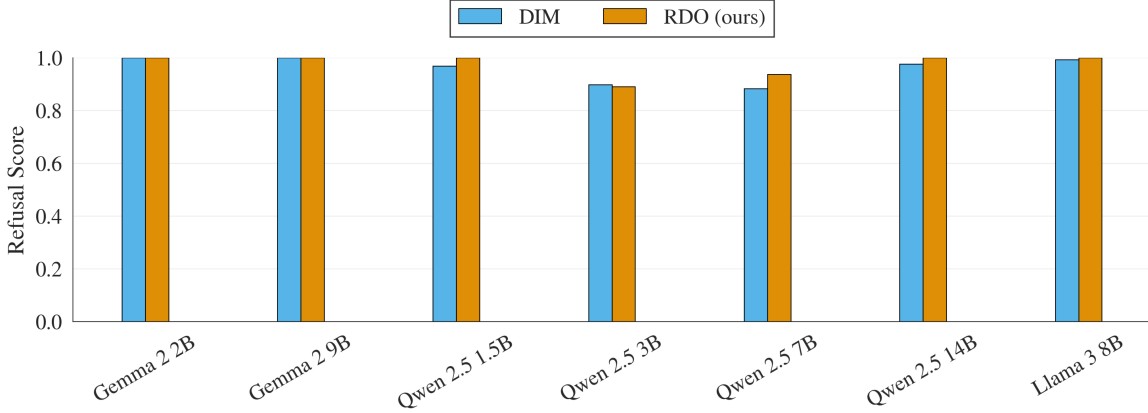

*Figure 10.* Refusal scores of different models on harmless instructions after activation addition that aims to induce refusal.

## B.2. Directional Ablation on Additional Datasets

For a more robust evaluation of how the RDO directions compare to DIM in terms of performance, we additionally evaluate directional ablation ASR on STRONGREJECT(Souly et al., 2024) and the SORRY-BENCH base dataset (Xie et al., 2025). We include baseline performance here without any intervention. The results are shown in Figure 11, Figure 12, and Figure 13, and confirm that our findings transfer across datasets.

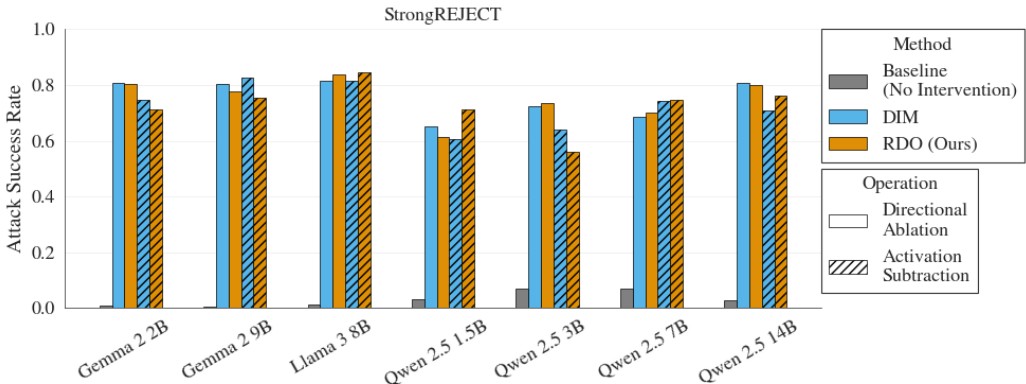

*Figure 11.* Attack success rates of refusal directions on STRONGREJECT.

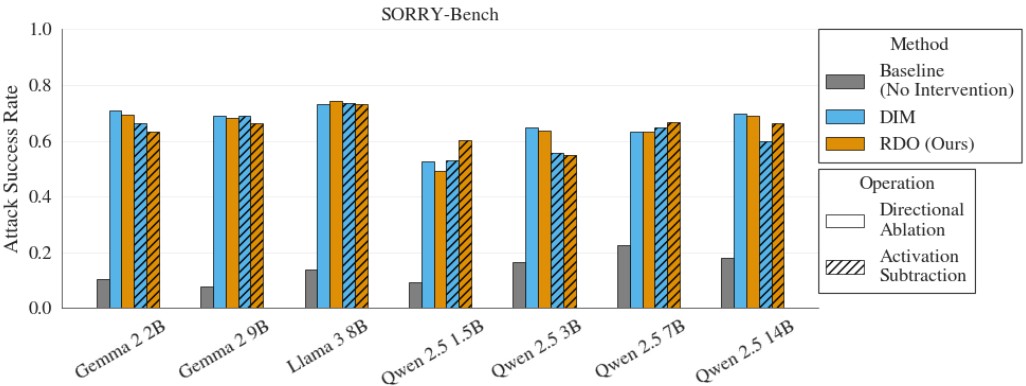

*Figure 12.* Attack success rates of refusal directions on SORRY-BENCH.

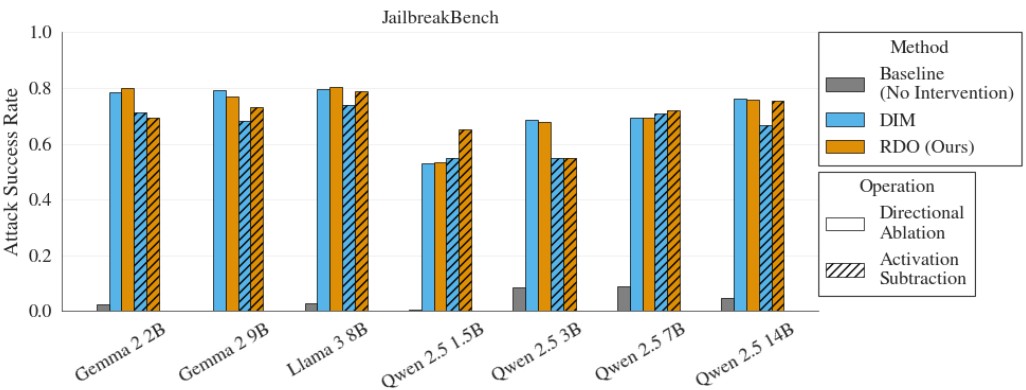

*Figure 13.* Attack success rates of refusal directions on JAILBREAKBENCH.

### B.3. Benchmarks

In Table 1 we showed that RDO directions have significantly lower side-effects on model performance as measured via the reduction in TRUTHFULQA score. Here, we show the change in benchmark scores for more benchmarks, specifically ARC (Clark et al., 2018), GSM8K (Cobbe et al., 2021), and MMLU (Hendrycks et al., 2020). In Table 4, we see that for most combinations of model and benchmark, the RDO direction has less impact on benchmark score, which confirms the effectiveness of our retain loss and that the RDO directions manipulate refusal more precisely. Finally, we summarize all results in Table 5.

*Table 4.* Sides effect measured by comparing model performance on benchmark datasets when ablating with either DIM or RDO. Ablating RDO has significantly lower side effects for most models and benchmarks.

| Dataset | Model | DIM | RDO | Change |
|---|---|---|---|---|
| TruthfulQA_MC2 | Gemma 2 2B | 8.0% | 4.4% | (-3.6%) |
| | Gemma 2 9B | 8.3% | 4.4% | (-3.9%) |
| | Llama 3 8B | 4.1% | 1.8% | (-2.3%) |
| | Qwen 2.5 1.5B | 3.6% | 2.6% | (-1.1%) |
| | Qwen 2.5 7B | 4.4% | 3.1% | (-1.2%) |
| | Qwen 2.5 14B | 7.5% | 2.5% | (-5.0%) |
| ARC_CHALLENGE | Gemma 2 2B | 0.2% | 0.2% | (-0.0%) |
| | Gemma 2 9B | 0.6% | 0.3% | (-0.3%) |
| | Llama 3 8B | 1.0% | 0.2% | (-0.9%) |
| | Qwen 2.5 1.5B | 0.7% | 0.3% | (-0.4%) |
| | Qwen 2.5 7B | 0.9% | 0.0% | (-0.9%) |
| | Qwen 2.5 14B | 1.5% | 0.6% | (-0.9%) |
| GSM8K | Gemma 2 2B | 0.6% | 0.4% | (-0.2%) |
| | Gemma 2 9B | 0.1% | 0.3% | (+0.2%) |
| | Llama 3 8B | 1.0% | 2.7% | (+1.7%) |
| | Qwen 2.5 1.5B | 1.1% | 0.0% | (-1.1%) |
| | Qwen 2.5 7B | 0.8% | 0.7% | (-0.2%) |
| | Qwen 2.5 14B | 1.5% | 0.5% | (-1.1%) |
| MMLU | Gemma 2 2B | 0.3% | 1.2% | (+1.0%) |
| | Gemma 2 9B | 1.3% | 0.1% | (-1.2%) |
| | Llama 3 8B | 2.6% | 0.1% | (-2.5%) |
| | Qwen 2.5 1.5B | 1.5% | 0.5% | (-1.0%) |
| | Qwen 2.5 7B | 0.7% | 0.1% | (-0.6%) |
| | Qwen 2.5 14B | 1.2% | 0.3% | (-0.9%) |

*Table 5.* Comparing RDO and DIM in terms of jailbreaking effectiveness and how jailbreaking affects general capability benchmarks. Each cell in the jailbreaking section contain pairs of ASRs: the first for directional ablation and the second for activation subtraction. The values in the general capability section are computed under directional ablation of the respective directions.

| | Jailbreaking | | | General Capability | | | |
|---|---|---|---|---|---|---|---|
| | **JailbreakBench** ASR ↑ | **StrongREJECT** ASR ↑ | **SORRY-Bench** ASR ↑ | **MMLU** Acc ↑ | **ARC-C** Acc ↑ | **GSM8K** Acc ↑ | **TruthfulQA** Acc ↑ |
| GEMMA 2 2B | 2.5 | 1.0 | 10.4 | 30.5 | 42.7 | 52.3 | 55.8 |
| DIM | 78.6 / **71.2** | **80.7 / 74.6** | **71.0 / 66.3** | **30.3** | 42.5 | **52.9** | 47.8 |
| RDO | **79.9** / 69.5 | 80.5 / 71.4 | 69.5 / 63.3 | 29.3 | **42.8** | 52.7 | **51.4** |
| GEMMA 2 9B | 0.3 | 0.6 | 7.8 | 34.8 | 52.6 | 74.7 | 61.1 |
| DIM | **79.1** / 68.3 | **80.2 / 82.7** | **69.1 / 69.0** | **36.1** | **53.2** | 74.6 | 52.8 |
| RDO | 76.8 / **73.0** | 77.7 / 75.4 | 68.1 / 66.3 | 34.7 | 52.2 | **75.0** | **56.7** |
| LLAMA 3 8B | 2.9 | 1.2 | 14.0 | 58.1 | 48.6 | 63.5 | 52.8 |
| DIM | 79.7 / 74.1 | 81.5 / 81.4 | 73.0 / **73.4** | 55.5 | 47.6 | 62.5 | 48.7 |
| RDO | **80.3 / 79.0** | **83.8 / 84.7** | **74.4** / 73.2 | **58.2** | **48.5** | **66.2** | **51.0** |
| QWEN 2.5 1.5B | 0.4 | 3.1 | 9.2 | 58.3 | 38.5 | 56.4 | 46.5 |
| DIM | 53.1 / 55.0 | **65.2** / 60.4 | **52.5** / 53.0 | 56.8 | 37.8 | 55.3 | 42.9 |
| RDO | **53.5 / 65.2** | 61.3 / **71.3** | 49.1 / **60.1** | **57.8** | **38.7** | **56.4** | **44.0** |
| QWEN 2.5 3B | 8.4 | 6.9 | 16.7 | 64.6 | 42.8 | 61.2 | 57.2 |
| DIM | **68.5 / 54.9** | 72.2 / **64.1** | **64.8 / 55.6** | 64.5 | **42.7** | **60.3** | 54.2 |
| RDO | 67.6 / 54.8 | **73.6** / 55.9 | 63.6 / 55.0 | **64.7** | 41.6 | 59.5 | **54.5** |
| QWEN 2.5 7B | 9.1 | 7.1 | 22.7 | 68.8 | 45.6 | 77.9 | 63.1 |
| DIM | 69.2 / 71.0 | 68.8 / 74.1 | 63.1 / 64.8 | 68.1 | **46.5** | 77.0 | 58.7 |
| RDO | **69.3 / 72.0** | **70.0 / 74.8** | **63.4 / 66.7** | **68.7** | 45.6 | **77.2** | **60.0** |
| QWEN 2.5 14B | 4.9 | 2.9 | 17.9 | 76.9 | 52.8 | 81.7 | 70.8 |
| DIM | **76.2** / 66.6 | **80.8** / 70.7 | **69.6 / 60.0** | 75.7 | 51.4 | 80.2 | 63.4 |
| RDO | 75.7 / **75.6** | 79.9 / **76.2** | 69.0 / **66.2** | **76.6** | **52.2** | **81.3** | **67.9** |

# C. Ablation Studies

We conduct ablation studies to determine the importance of the three losses in our RDO algorithm.

## C.1. Addition and Ablation Loss

We first study how the addition and ablation loss should be balanced. We experiment with the Llama-3-8B-Instruct model and range the loss weights $\lambda_{abl}$ and $\lambda_{add}$ from 0 to 1, setting $\lambda_{abl} = 1 - \lambda_{add}$ to balance the weights. We then evaluate attack success rates for both directional ablation and activation subtraction interventions. Figure 14 shows that both loss components are essential for optimal performance. ASR is similar across the 0.2–0.8 weight range, where both methods maintain consistently high attack success rates above 80%, and choosing only one of the losses reduces performance significantly. This finding indicates that while including both loss terms is critical, the precise weight allocation within this range has minimal impact on effectiveness. This robustness simplifies hyperparameter tuning in practice, as practitioners can select any weight configuration within this range without substantially affecting performance.

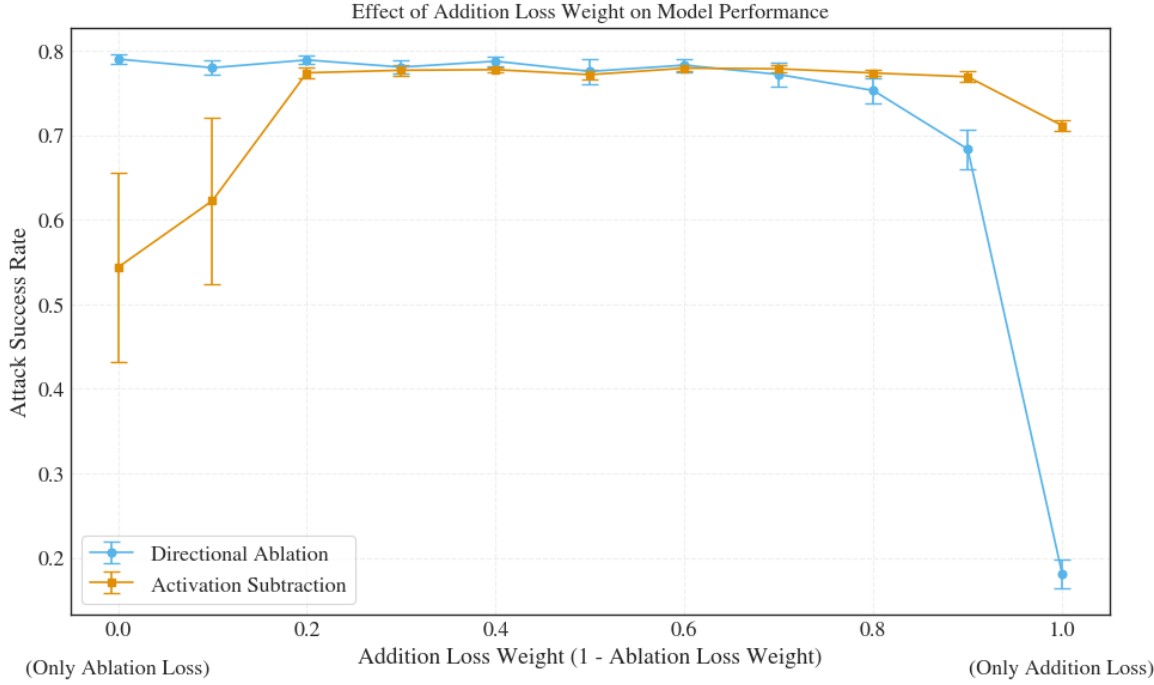

*Figure 14.* Ablation study of loss weights $\lambda_{abl}$ and $\lambda_{add}$ for Llama-3-8B-Instruct. We compare attack success rates for directional ablation and activation subtraction across different weight balances ($\lambda_{abl} = 1 - \lambda_{add}$). Both methods perform well in the 0.2–0.8 range, with severe degradation at extremes, particularly for directional ablation using only the addition loss (20% ASR at $\lambda_{add} = 1$).

## C.2. Retain Loss

Regarding the impact of using the retain loss to minimize side-effects when intervening with our refusal directions, we conduct an ablation study of the retain loss weight $\lambda_{ret}$ for Qwen2.5-3B-Instruct. We fix the ablation and addition loss weights to their default values (see Table 3) and systematically vary $\lambda_{ret}$. Figure 15 presents a Pareto analysis plotting ASR under directional ablation against the average reduction in benchmark performance that results from directional ablation. In more detail, the x-axis represents the average change in benchmark scores when intervening with the refusal direction across GSM8K, MMLU, ARC-Challenge, and TruthfulQA relative to the model's baseline performance (no intervention), while the y-axis shows the corresponding JAILBREAKBENCH ASR under directional ablation. The ideal refusal direction would maximize ASR while maintaining or improving benchmark performance (e.g., by preventing the model from inappropriately refusing legitimate questions).

For this specific model, retain weights up to $\lambda_{ret} = 4$ increasingly reduce side-effects on benchmark performance with only marginal ASR reduction. However, beyond this threshold, the ASR drops drastically, indicating that excessive weighting of

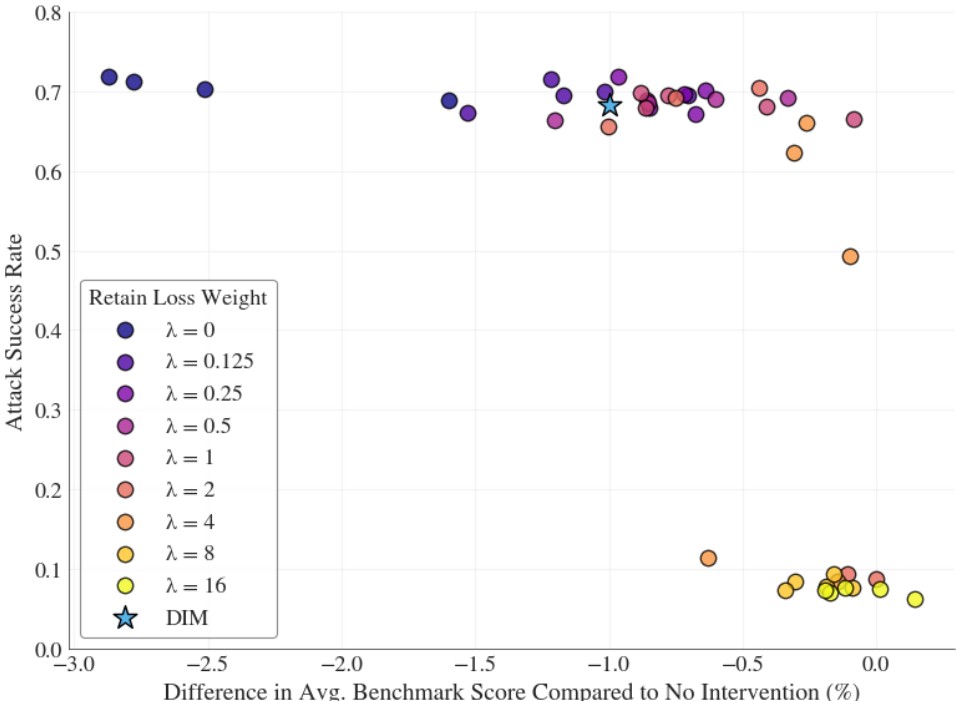

*Figure 15.* Ablation study of retain loss weight $\lambda_{\text{retain}}$ for Qwen2.5-3B-Instruct. We plot JailbreakBench ASR versus benchmark perfor-mance change (x-axis, averaged over GSM8K, MMLU, ARC-Challenge, TruthfulQA) when intervening with the direction compared to baseline. Higher retain weights improve benchmark preservation with minimal ASR loss up to $\lambda_{\text{retain}} = 4$, after which ASR drops significantly. Multiple hyperparameter choices Pareto-dominate the DIM baseline, demonstrating robust improvements over current state-of-the-art.

the retain loss impedes the learning of effective refusal directions.

Importantly, we observe that multiple hyperparameter configurations achieve Pareto dominance over the DIM baseline across both metrics. This demonstrates that our method provides robust improvements over the current state-of-the-art, rather than gains limited to specific hyperparameter choices.

## D. Flexibility of Algorithms

We find that DIM can struggle to identify refusal directions with sufficiently low side-effects (according to the heuristic used by Arditi et al. (2024). Figure 16 visualizes the effectiveness of the direction selection algorithm from Arditi et al. (2024) for DIM directions in the Qwen 2.5 7B model. Among the evaluated token and layer pairs, only one direction is found to be effective for both inducing refusal through activation addition and maintaining low side effects. Transparent data points indicate (layer, token) combinations that were filtered out due to their inability to induce refusal reliably. Additionally, the red line represents the KL-divergence threshold, used to estimate potential side effects of directional ablation on harmless instructions.

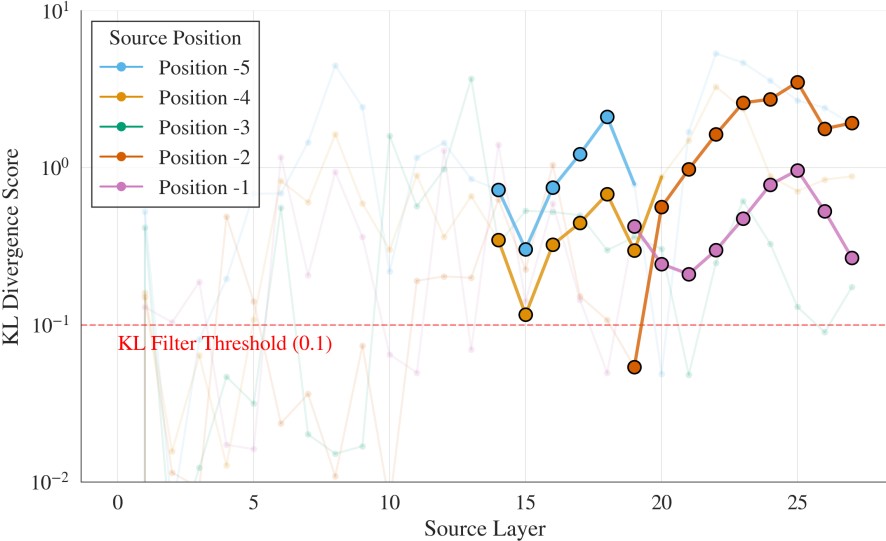

*Figure 16.* Analysis of the selection direction algorithm from Arditi et al. (2024) for the DIM directions of Qwen 2.5 7B. Among the token and layer combinations, only a single direction is identified as viable for both inducing refusal via activation addition and having low side-effects. Transparent points represent (layer, token) pairs that are filtered out because of ineffectiveness in inducing refusal. The red line indicates the KL-divergence threshold used to estimate potential side-effects of directional ablation on harmless instructions.

# E. Extended Results for Refusal Cones

In Figure 17 we show the refusal cones for the Gemma model family.

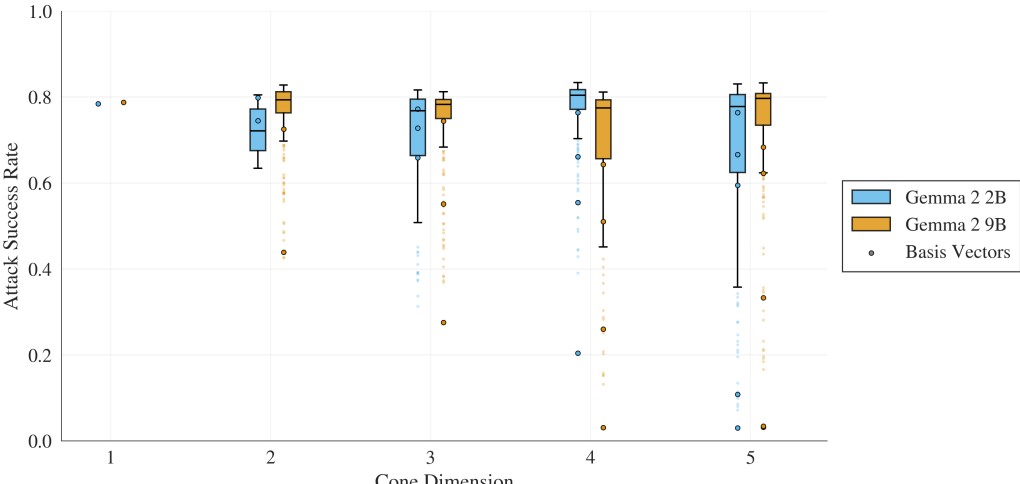

*Figure 17.* Attack success rates in refusal cones of different dimensions for the Gemma 2 model family. We observe that for the Gemma 2 2B the lower bounds start to degrade significantly for dimension 5.

In Figure 18 we measure the performance of the refusal cones in the Qwen 2.5 model family for activation addition. The models tend to support higher dimensional cones compared to Figure 4, revealing that inducing refusal is significantly easier than disabling refusal via directional ablation. Notably, most directions even in high-dimensional cones remain effective at inducing refusal responses.

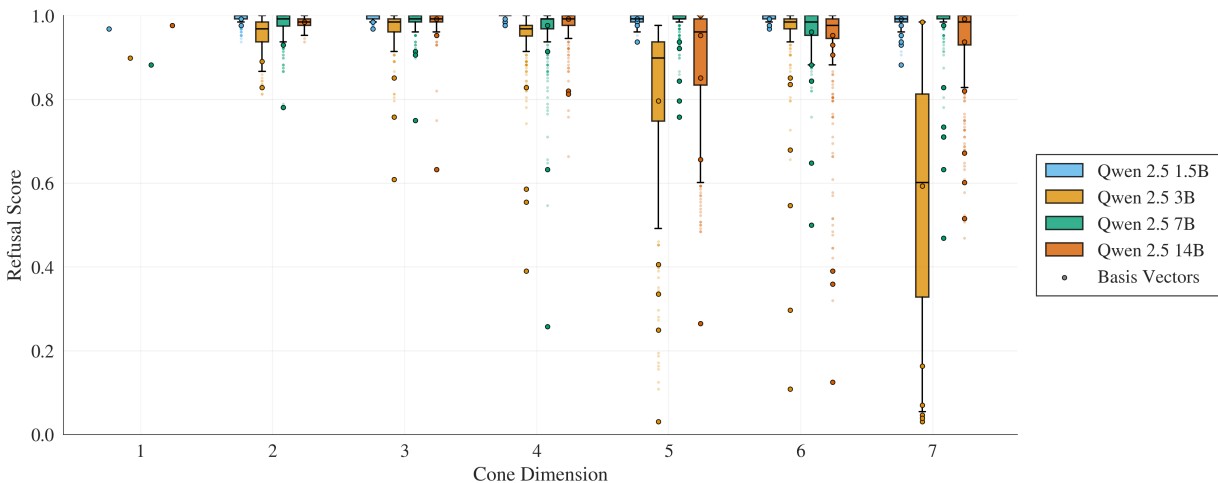

*Figure 18.* Using refusal cones to induce refusal across various Qwen 2.5 models with different dimensions. We observe that inducing refusal is generally easier than executing an attack. In this setting, nearly all dimensions maintain strong performance in eliciting refusal responses.

Figure 19 examines the attack success rate when sampling multiple vectors from various $N$-dimensional refusal cones and selecting the best-performing sample per prompt for Gemma 2, 2B. We observe that ASR improves with increasing cone dimensionality but plateaus at four dimensions, suggesting that higher-dimensional cones provide an advantage over single-direction manipulation by capturing complementary mechanisms. The plateau likely results from the model's inability to encode higher-dimensional refusal cones, a hypothesis further supported by Figure 17.

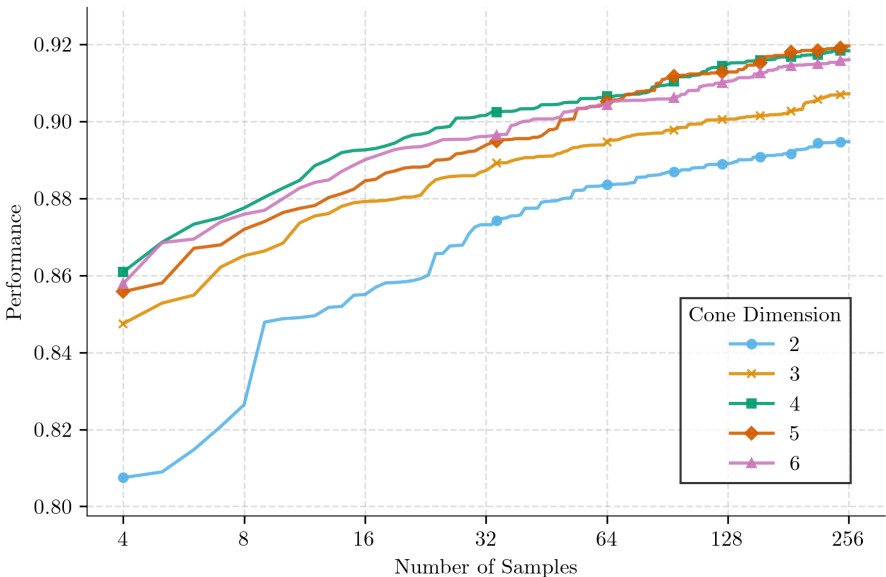

*Figure 19.* Attack success rates when sampling vectors from the N-dimensional refusal cones and selecting the best-performing sample per prompt for Gemma 2 2B. ASR increases with cone dimensionality but plateaus at four dimensions, suggesting that higher-dimensional cones provide an advantage over single-direction manipulation by capturing complementary mechanisms. The plateau likely arises because Algorithm 2 cannot find an additional basis vector that preserves the refusal properties in the cone, suggesting that the model does not support a cone of this dimension. Figure 17 also provides evidence for this claim.

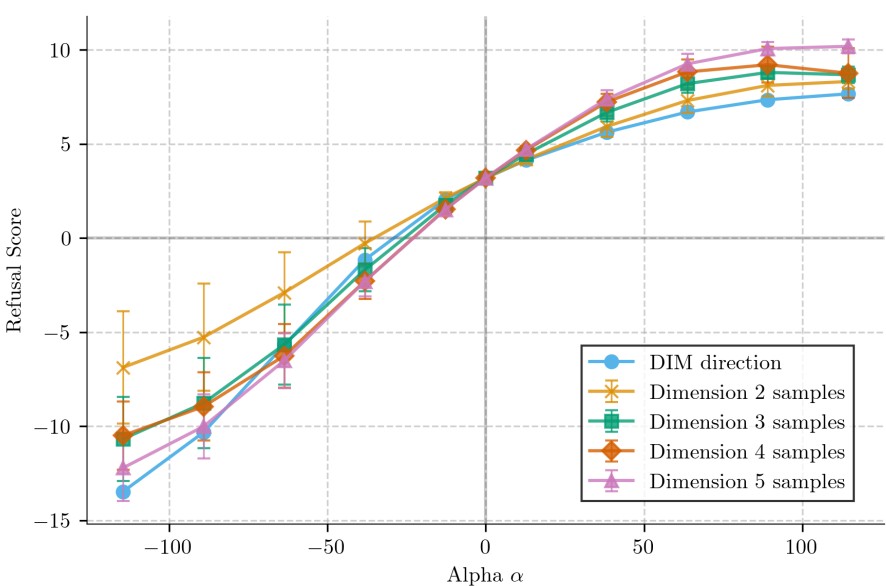

*Figure 20.* Refusal scores of refusal vectors sampled from Gemma 2 2B refusal cones compared to the DIM direction when scaling the norm of the added direction $\alpha$ for the activation addition intervention. The refusal score is the heuristic from Arditi et al. (2024) here, and we compute it on 64 harmful validation instructions, with mean and standard deviation over 64 samples per alpha.

# F. Orthogonal Refusal Directions

In Figure 6 we showed how the orthogonal $RDO_\perp$ interacts with the DIM direction. Here, we show the result for two additional models, on a larger dataset (SORRY-BENCH). Figure 22 supports that the DIM direction is greatly influenced by ablating orthogonal refusal directions, supporting our claims in Section 6.1.

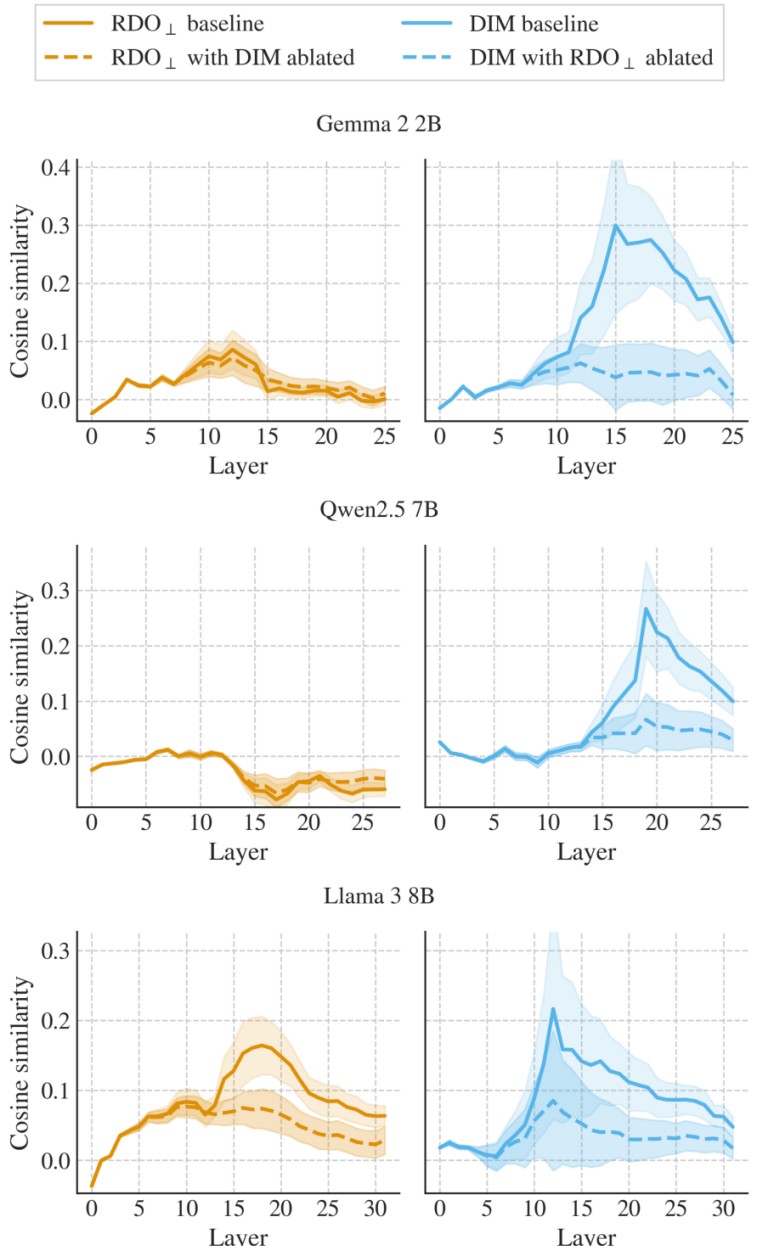

*Figure 21.* Interaction of orthogonal refusal directions directions under directional ablation, measured in terms of cosine similarity to model activations at the last token position on harmful instructions from SORRY-BENCH, across different models.

# G. Over-refusal

We evaluate the trade-off between over-refusal and safety by analyzing activation addition of the RDO and DIM directions across different strength configurations for the Gemma 2 2B model. Figure 22 shows results for activation addition strengths ($\alpha$) ranging from 0 to $||v||_2$, where we compute refusal scores on two distinct datasets: harmful instructions from SORRY-BENCH and harmless instructions from XSTEST (Röttger et al., 2023). Higher scores on SORRY-BENCH indicate safer responses to genuinely harmful prompts, while lower scores on XSTEST indicate reduced over-refusal on benign inputs that are designed to measure inappropriate refusal behavior. Our method consistently provides better or equivalent trade-offs compared to DIM across all strength configurations, suggesting that RDO directions may be more suited for increasing safety at the same rate of over-refusal compared to DIM.

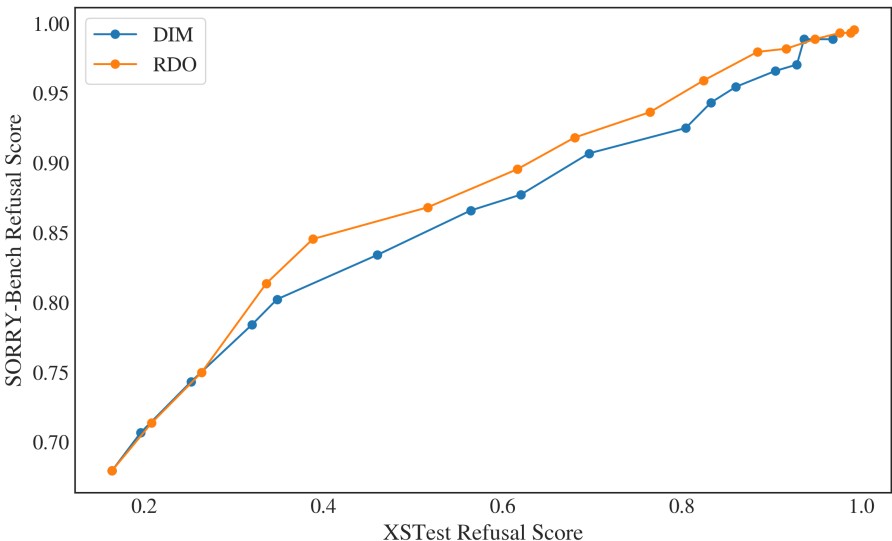

*Figure 22.* Trade-off between over-refusal and safety for different activation addition strengths of our refusal directions.

# H. Assets

In the following, we show the licenses for all the assets we used in this work: different models from Table 6 and the datasets that we use for evaluation and training; see Table 7.

## H.1. Models

*Table 6.* The list of models used in this work.

| Model | Source | Accessed via | License |
|---|---|---|---|
| **Qwen 2.5 {1.5B, 7B, 14B}** | Yang et al. (2024) | Link | Apache 2.0 License |
| **Qwen 2.5 {3B}** | Yang et al. (2024) | Link | Qwen Research License |
| **Gemma 2 2B** | Team et al. (2024) | Link | Apache 2.0 License |
| **Gemma 2 9B** | Team et al. (2024) | Link | Gemma Terms of Use |
| **Llama-3 8B** | Dubey et al. (2024) | Link | Meta Llama 3 Community License |
| **StrongREJECT Judge** | Souly et al. (2024) | Link | MIT License |

## H.2. Datasets

*Table 7.* The list of datasets used in this work.

| Dataset | Source | Accessed via | License |
|---|---|---|---|
| SALADBENCH | Li et al. (2024a) | Link | Apache License 2.0 |
| ALPACA | Taori et al. (2023) | Link | Apache License 2.0 |
| JAILBREAKBENCH | Chao et al. (2024) | Link | MIT License |
| STRONGREJECT | Souly et al. (2024) | Link | MIT License |
| SORRY-BENCH | Xie et al. (2025) | Link | Custom License |
| XSTEST | Röttger et al. (2023) | Link | CC-BY-4.0 |
| TRUTHFULQA | Lin et al. (2021) | Link | Apache License 2.0 |
| MMLU | Hendrycks et al. (2020) | Link | MIT License |
| ARC | Clark et al. (2018) | Link | CC-BY-SA-4.0 |
| GSM8K | Cobbe et al. (2021) | Link | MIT License |

