# OpenReview forum: "The Geometry of Refusal in Large Language Models: Concept Cones and Representational Independence"
_ICML.cc/2025/Conference — ICML 2025 poster_

### Official Review · Reviewer_NvcZ · 2025-03-01

**Overall Recommendation:** 3

**Summary:**

This paper explores the refusal behavior of LLMs and introduces a gradient-based algorithm called RDO to identify refusal directions $r$ in the activation space. These directions can shift the model’s behavior towards either refusal or acceptance. The authors design three distinct loss terms to help identify directions that can trigger refusal behavior while ensuring that other unrelated behaviors remain unaffected when adjusting refusal directions. Additionally, they extend their analysis to an N-dimensional polyhedral cone of refusal directions rather than a single vector $r$. The paper further examines whether the orthogonal refusal directions identified by RDO are independent and can be manipulated by modifying the input to the model. Experimental results reveal several key insights into the refusal behavior of LLMs, enhancing the understanding of this phenomenon.

**Claims And Evidence:**

The paper presents three main claims, each supported by a combination of theoretical arguments and empirical evidence. Overall, the evidence is generally convincing.

**1. Gradient-based representation engineering can identify refusal directions.**
- The authors introduce the Refusal Direction Optimization (RDO) method, which uses a gradient-based approach to identify vectors in the activation space that control refusal behavior.
- They validate this claim by demonstrating that applying these vectors can reliably increase the probability of refusal (scaling property) and that projecting out these directions allows the model to respond to harmful prompts while retaining behavior on harmless prompts (projection property).
- The use of cross-entropy loss for training these directions and the retain loss based on KL divergence to prevent interference with other model behaviors provides a robust framework for evaluating this claim.

**2. Refusal behavior is governed by multi-dimensional cones rather than a single direction.**
- The authors did experiments that measure ASR across cones with increasing dimensionality. The observation that higher-dimensional cones consistently mediate refusal behavior across different models suggests that refusal is indeed encoded in a multi-dimensional structure.
- The use of orthonormal basis vectors ensures that the identified directions are non-overlapping, which further strengthens the evidence for multi-dimensional control of refusal.

**3. Orthogonal directions in the cone are not just geometrically independent but also causally independent**
- The experiment uses cosine similarity to test the interaction between orthogonal directions show that ablating one direction does not influence the other, demonstrating the causal independence of different directions.

**Essential References Not Discussed:**

No.

**Ethical Review Concerns:**

No.

**Experimental Designs Or Analyses:**

I examined the methodology behind the refusal direction optimization experiments, focusing primarily on the Attack Success Rate (ASR) metric and the directional ablation approach used to measure changes in refusal behavior. This setup is sound because it isolates the influence of specific directions in the activation space. However, relying on a single metric can overlook nuances in how robust or consistent those directions are across diverse prompts. Additionally, while the authors’ sampling strategy provides a reasonable set of prompts, the limited number of harmful examples may restrict the generalizability of the findings.

The layer-wise analysis of orthogonal directions is a valuable step toward understanding deeper interactions in the model, but it is relatively narrow in scope, concentrating on one model architecture (Gemma 2 2B) and a small set of prompts. Broadening the experiments to include multiple models and larger, more varied prompt sets could improve the robustness of the conclusions.

**Methods And Evaluation Criteria:**

The methods proposed in the paper, particularly RDO and its extension to multi-dimensional cones, are well-designed for probing and controlling refusal behavior in LLMs. The use of a gradient-based approach to identify refusal directions and the reliance on orthonormal bases to define multi-dimensional cones are reasonable choices given the complexity of activation spaces in LLMs. The evaluation strategy, which primarily hinges on the Attack Success Rate (ASR) and cosine similarity measures, provides an empirical basis for the claims made. These choices align well with the paper’s objectives of understanding and isolating the mechanisms of refusal behavior.

However, there are several limitations that could impact the validity and generalizability of the findings. Firstly, they rely on ASR as the main metric to evaluate the quality of the cones, which might be limited. Could there be alternative or complementary metrics that can better capture, e.g., the robustness of the cones? Additionally, the use of cosine similarity to infer causal independence between orthogonal directions might be insufficient. Cosine similarity captures only linear relationships, potentially overlooking complex, non-linear interactions between directions in the activation space. Furthermore, the experiments on orthogonality focus on a small subset of harmful prompts and a single LLM architecture (Gemma 2 2B). This raises concerns about the robustness of the results across different model architectures and prompt types. It would be beneficial to investigate larger models and datasets. Addressing these limitations through a broader set of metrics—such as mutual information or causal inference techniques—and expanding the evaluation to include a wider range of prompts and models could significantly strengthen the paper’s contributions.

**Other Comments Or Suggestions:**

No.

**Other Strengths And Weaknesses:**

No. Please see the previous sections.

**Questions For Authors:**

1. Could the authors provide more insights or motivations regarding the importance of identifying refusal directions for LLMs? What specific objectives can be achieved using these refusal directions? A deeper understanding of these aspects could help clarify the scope of the paper.
2. In Table 1, higher values indicate better performance. Why does RDO perform significantly worse than the baseline?
3. In Figure 2, it appears that the performance of RDO is not substantially better than that of DIM. What might be the reasons for this?

**Relation To Broader Scientific Literature:**

The paper’s core contribution—using gradient-based representation engineering to identify and manipulate “refusal directions” in LLMs—builds upon several lines of existing research in interpretability and controllability. Recent works have investigated how specific concepts, such as sentiment or toxicity, can be localized in a model’s latent space and how targeted interventions (e.g., projecting out specific directions) can alter generation behaviors. The authors’ approach extends these ideas by focusing on refusal behaviors, a safety-critical function in many conversational and policy-driven systems.

By introducing a gradient-based approach rather than relying solely on paired prompts or heuristic methods, the paper also connects to the broader endeavor of fine-grained model steering. It strengthens a growing body of evidence that language models’ internal representations can be both discovered and selectively edited to meet specific policy or safety goals.

**Theoretical Claims:**

The proof for the scaling property is straightforward and relies on the assumption that scaling a direction within the activation space should proportionally increase refusal behavior. This proof appears correct, as it logically follows from the gradient-based optimization objective designed to maximize refusal probability. The mathematical argument, which demonstrates that a higher scaling factor leads to an increased likelihood of refusal, is both clear and valid within the context of the assumptions made.

The projection property proof is also reasonable, showing that projecting out the identified directions should enable the model to respond to harmful prompts while maintaining behavior on harmless ones. The proof leverages the orthogonality of the directions to argue that this ablation selectively removes the influence of refusal-related factors. While the proof is correct, given the orthogonality assumption, one potential issue is that it does not fully address the possibility of non-linear interactions between directions that might influence the model’s behavior indirectly.

---

> ### Author Rebuttal · Authors · 2025-04-01
>
> We thank the reviewer for the time and effort to review our manuscript which helps us to improve.
>
> ### Cosine similarity captures linear relationships. RepInd also catches non-linear interactions.
> If the reviewer means that the orthogonality of the cone basis vectors from section 5 is not enough, we fully agree! This is exactly why we investigate the representational independence in section 6.
>
> In case the reviewer questions our representational independence in mitigating non-linear effects this might be due to a missing equation. We noticed that we forgot to specify the definition of the ablated $\hat{x}$.
>
> $$
> \hat{x}\_{abl}^{(l)} =  [\hat{x}\_{\text{abl}}^{(l-1)} +  f_{[l]}(\hat{x}\_{\text{abl}}^{(l-1)})]\_\text{abl}
> $$
>
> where $f_{[l]}$ is the model's transformation at layer $l$.
>
> This means, that for rep. ind. we enforce the cosine similarity to be equal to the no-intervention baseline for every layer while propagating the ablated representation and ablating again. Intuitively, any non-linear effects would make it impossible to have the equality of cosine similarities of the subsequent layers. We will provide a simple proof by induction for the camera-ready version of the manuscript if the reviewer thinks this will bring clarity.
> To argue for causal independence, we would need to show that the cosine similarity to a concept is what drives the model prediction. Since the readout of standard transformers is not linear, we cannot prove this unfortunately. However, there exists some empirical evidence in the mechanistic interpretability field (Elhage et al., Geva et al., Yousefi et al.) that the directional information (captured by cosine similarity) is what greatly drives the model's readout.
> We will provide an extensive discussion of this topic in section 6 and thank the reviewer for the question as discussing it will improve our manuscript!
>
> ### Alternative metrics apart from ASR
> We provide alternative metrics to evaluate our refusal cones beyond the ASR. While ASR offers the most comprehensive assessment of the cone directions' effectiveness for model manipulation, it is computationally expensive. A more efficient alternative is the refusal metric proposed by Arditi et al. (2024). Using this metric, we show in [Figure 14](https://figshare.com/s/4ab2ec422f6bd0262b30) that the cone directions fulfill the monotonic scaling property we required from our refusal directions.
>
> Besides this, we now include over-refusal as an additional way to examine the performance of RDO directions, see response Over-Refusal to reviewer 4YKU.
> If the reviewer has another idea on what we should evaluate or could give us any pointers, we would be happy to include more evidence in the next reply.
>
> ### More data and models
> While we used the same test dataset as related work (Arditi et al., 2024), we agree with the reviewer that we could add more to convincingly validate our claims. We now include the datasets of StrongREJECT and SORRY-Bench, where our results behave similarly to JailbreakBench, strengthening our findings, see [Figures 21-23](https://figshare.com/s/4ab2ec422f6bd0262b30).
> Further, we added more models and data for the orthogonality / representational independence experiments, see [Figure 26, 28](https://figshare.com/s/4ab2ec422f6bd0262b30).
>
> ### Motivation for Refusal Directions (Q1)
> Due to writing constraints, we kindly ask the reviewer to read our **Applications** response to reviewer 5cSG. Apart from safety-relevant topics, this work further shows that RDO & RCO are suitable techniques to identify directions for precise representation steering which is currently a promising technique for LLM understanding.
>
> ### Questions 2 & 3
> In Table 1, the term 'Baseline' was initially unclear—we now clarify it refers to 'No intervention'. While RDO leads to significantly fewer side effects than DIM, it still introduces some compared to no intervention. Since LLMs are highly optimized, any intervention can reduce coherence. Additionally, benchmarks include sensitive questions, where safety interventions can yield differing outputs.
>
> The relative performance has several explanations: (1) DIM was used to generate labels, creating a performance ceiling; (2) The StrongREJECT judge assesses harmfulness, which also reflects model capability. Outperforming DIM was not our goal—RDO serves as a foundation for RCO and the study of independent refusal directions. Figure 2 demonstrates that RDO achieves comparable or slightly better mediation performance with greater precision (fewer side effects). With further tuning, RDO could surpass DIM, but our focus is on proposing an alternative approach and analyzing its properties.
>
> ### Conclusion
> We again want to thank the reviewer for their insightful questions and remarks. The changes we applied to the paper improved our manuscript. We hope the reviewer agrees with us and would be happy to answer any further questions or concerns.

---

### Official Review · Reviewer_oMnW · 2025-03-12

**Overall Recommendation:** 4

**Summary:**

This paper investigates the refusal mechanism of LLMs with a gradient-based representation engineering method. They extract refusal directions with gradient optimization by transferring the two refusal direction requirements (addition and ablation) into two loss functions. Further, they give several novel findings: 1) there are multi-dimensional concept cones mediating refusal, and 2) we can find multiple refusal directions that are independent from each other. The research provides a foundation for future studies to understand and improve the robustness and reliability of LLMs in handling adversarially crafted inputs.

## update after rebuttal
This paper presents a thorough investigation and offers interesting findings on the refusal mechanism of LLMs, which is a highly relevant topic. I continue to support acceptance.

**Claims And Evidence:**

The claims in the paper are well-supported by experiments.

**Essential References Not Discussed:**

NA

**Experimental Designs Or Analyses:**

The experimental design effectively addresses the research questions, and the analysis presents very novel findings.

**Methods And Evaluation Criteria:**

The evaluation benchmark used is JailbreakBench, which, while generally suitable, has shown biases in specific areas [2]. Including additional datasets such as AdvBench [1] or SorryBench [2] could provide a more comprehensive evaluation. Have you considered using other datasets?

[1] Zou, A., Wang, Z., Carlini, N., Nasr, M., Kolter, J. Z., & Fredrikson, M. (2023). Universal and transferable adversarial attacks on aligned language models. arXiv preprint arXiv:2307.15043.

[2] Xie, T., Qi, X., Zeng, Y., Huang, Y., Sehwag, U. M., Huang, K., ... & Mittal, P. (2024). Sorry-bench: Systematically evaluating large language model safety refusal behaviors. arXiv preprint arXiv:2406.14598.

**Other Comments Or Suggestions:**

NA

**Other Strengths And Weaknesses:**

NA

**Questions For Authors:**

In the direction addition and ablation experiments, do you focus on a specific layer or do you apply modifications across all layers?

**Relation To Broader Scientific Literature:**

The paper presents novel insights into the refusal mechanisms of LLMs, enhancing our understanding of AI safety and encouraging further research into the robustness and reliability of LLMs.

**Theoretical Claims:**

NA

---

> ### Author Rebuttal · Authors · 2025-04-01
>
> We thank the reviewer for their valuable time for the review, the kind words, and the positive assessment.
>
>
> ### Additional Data
> We agree that we should increase the number of datasets we use for our evaluations. We thank the reviewer for the pointer towards SORRY-Bench and decided to evaluate our method on their base set of 440 harmful instructions. AdvBench is in our training data since it is part of SALAD-Bench and we therefore cannot use it for evaluation. As an alternative, we further evaluate on StrongREJECT.
>
> We show the results in [Figures 21-23](https://figshare.com/s/4ab2ec422f6bd0262b30) for our three datasets. We observe that the results behave very similar to JailbreakBench which shows the generalizability of our findings and improves our manuscript. For the camera ready version we changed all figures to include the evaluation of all datasets together, instead of having plots for each dataset. If the reviewer suggests a different way to display the results though, we are happy to adjust.
>
>
> ### Intervention positions
> We apply the ablation operation across all layers and the activation addition on a single layer. This corresponds to the previous work by Arditi et al. in *Refusal is Mediated by a Single Direction*. We further always train the RDO directions and cones in the same layer as the DIM direction for a fair comparison. Future work could look at studying refusal directions across all layers. However, if the reviewer wants to see results for a different approach we are happy to investigate.
>
> ### Conclusion
> We again thank the reviewer for their time and effort. The additional data improves the significance of our findings and strengthens this work. We further added more experiments, other metrics, and evidence for causal independence during this rebuttal phase and detailed the changes in the responses to the other respective reviewers. We believe that the rebuttal further improved our work and we hope that the reviewer agrees with us.

---

> > ### Comment · Reviewer_oMnW · 2025-04-04
> >
> > Thanks for the response. The additional experiments address all my concerns.
> >
> > One additional question: Is it possible that independent refusal directions could represent different types of harmfulness, e.g., violence, privacy violation, etc...

---

> > > ### Author Response · Authors · 2025-04-08
> > >
> > > Great question! We were thinking along the same lines: since, for example, privacy violations and harmful content typically trigger the standard "As a large language model..." refusal during RLHF training, the model may form different internal concepts that lead to those refusals. However, answering this question empirically might be difficult since, e.g., interpreting these directions with SAE features is likely unsuccessful because it is reported in related work that they don’t perform particularly well in slightly out-of-distribution settings. A follow-up work, however, that thoroughly interprets these directions would be very interesting and we add this as an outlook at the end of our manuscript.
> > >
> > > We again want to thank the reviewer for their time and are grateful for the feedback and pointers to provide more evidence for our method!

---

### Official Review · Reviewer_4YKU · 2025-03-12

**Overall Recommendation:** 4

**Summary:**

This work challenges the notion that refusal behavior in language models is mediated by a single direction. Previous research suggested that by ablating the activation strength along a specific direction, LMs could be made to refuse more or less often. The authors introduce a novel approach called Refusal Direction Optimization (RDO), which identifies optimal refusal directions while minimizing unintended effects on unrelated inputs. Their experiments demonstrate that RDO outperforms traditional methods that rely on contrasting prompts. Importantly, the authors show that multiple independent refusal directions can be found within the same model, suggesting that refusal mechanisms are more complex than previously understood.

**Claims And Evidence:**

The primary claims in this paper appear supported by the provided evidence with the exception of the overall performance section. I discuss this concern later in my review.

**Essential References Not Discussed:**

I am unaware of any obvious omissions that aren't concurrent work.

**Experimental Designs Or Analyses:**

I find the evidence supporting the claim that multiple refusal vectors can be identified to be convincing. However, the experimental design omits important measurements, making it difficult to assess how effectively their proposed techniques improve upon the baseline.

**Overall Performance**: Studying the effects on overall performance is an important consideration when introducing a new steering method—an aspect often neglected in previous works. However, measuring solely on TruthfulQA (TQA) provides a relatively narrow assessment of overall performance. My understanding is that the authors selected TQA primarily because this is a benchmark where the DIM baseline struggled. Beyond TQA being an insufficient evaluation of overall performance, selecting this benchmark solely based on the baseline's poor performance could lead skeptical readers to suspect cherry picking. I recommend that the authors include additional commonly studied benchmarks that measure broader capabilities, such as MMLU and GSM8K (though these specific benchmarks aren't strictly necessary).

Another important metric, especially if the paper aims to demonstrate the practicality of the technique, is over-refusal. Benchmarks like XStest and WildGuard are commonly used for this purpose. It would be valuable to investigate whether different refusal directions exhibit varying degrees of over-refusal.

**Baseline Improvements**: I was unable to find the baseline ASR without any steering applied. How significantly does steering improve ASR compared to not steering at all? This information seems particularly relevant given that my interpretation of Figure 2 suggests ASR remains above 50% even with steering applied.

**Methods And Evaluation Criteria:**

They do make sense. The ability to steer model activations during inference is a plausibly effective approach for reducing jailbreak attack success rates (ASR). The theoretical foundation behind this approach is sound, as manipulating the model's internal representations could intercept harmful outputs before they manifest. The authors' techniques build logically on previous work in this domain while addressing some of its limitations.

**Other Comments Or Suggestions:**

NA

**Other Strengths And Weaknesses:**

NA

**Questions For Authors:**

NA

**Relation To Broader Scientific Literature:**

This work challenges the notion that concepts of relevance to AI safety, such as refusal, are mediated by a single direction. The authors make their case via introducing a new direction identification technique.

**Theoretical Claims:**

I did not verify proofs. My interpretation is that the emperical evidence is strong.

---

> ### Author Rebuttal · Authors · 2025-04-01
>
> We thank the reviewer for their time and effort to examine our work. Below, we address the reviewers points.
>
> ### Overall performance
> We agree with the reviewer that our previous side-effect evaluation should be extended. We now provide results for the benchmarks: Arc Challenge, GSM8K, MMLU, and TruthfulQA (MC2), see [Table 5]((https://figshare.com/s/4ab2ec422f6bd0262b30)). While the improvement is not as strictly pronounced as for the Truthful QA, we observe that RDO performs better in the majority of the cases and provides on average a 24% reduction of error. We added these to the camera-ready appendix, linked to it from the main body, and adapted the framing in this paragraph to account for the overall improvement.
>
> ### Baseline Improvements
> We now provide the No-intervention baseline for the plot in Figure 2, see [Figures 21-23](https://figshare.com/s/4ab2ec422f6bd0262b30) for the respective datasets. It is important to note, that here in the experiment we show the attack success rate while trying to jailbreak the model, thus higher is better. We adjusted the text in the paper and the caption to make this more explicit. This, and our ablation study [Figures 18 & 19](https://figshare.com/s/4ab2ec422f6bd0262b30) shows we strictly improve the tradeoff between utility and ASR (higher is better) for our RDO direction compared to DIM as our direction Pareto-dominates DIM for many configurations. Due to space constraints, we kindly refer the reviewer to the 'Ablation studies ..' response to reviewer 5cSG for more details.
>
> ### Over-Refusal
> We acknowledge the important trade-off between safety and utility highlighted by the reviewer that is generally present for every safety method *compared* to no intervention. Our work primarily focuses on training refusal directions that are effective for model manipulation, specifically jailbreaking scenarios. Regarding over-refusal concerns,  adding refusal directions increases the probability of refusal generally, including over-refusal, as shown in Figure [Figure 25](https://figshare.com/s/4ab2ec422f6bd0262b30). Here, we add the refusal vector onto the representation with varying strength and plot the SORRY-Bench refusal score (higher is better) over the XSTest safe prompts refusal score (lower is better). We observe that RDO provides a better tradeoff as it refuses more harmful queries for the same rate of benign query refusal.
> Instead of increasing refusal, we can also *reduce* over-refusal by ablating the refusal direction. In [Figure 27](https://figshare.com/s/4ab2ec422f6bd0262b30) we show that results on safe instructions from the XSTest dataset. Here we observe that both approaches significantly reduce the already low over-refusal.
>
> While our current methodology focuses on identifying and understanding refusal mechanisms, rather than defending from adversarial attacks (which would have more direct implications on over-refusal), it could be extended to selectively reduce over-refusal without compromising refusal on harmful instructions by adjusting the loss functions. This could be achieved by using falsely refused instructions instead of harmful instructions in the ablation loss, and adding a retain loss for preserving refusal on harmful instructions. Though this specific application wasn't the focus of our current work, our framework can flexibly be repurposed for such interventions in future research.
>
> ### Conclusion
> We again thank the reviewer for the time, effort, and valuable feedback on our work. We believe that we can address all of the reviewer's concerns and hope the reviewer agrees with us. Further, we are happy to address any remaining questions!

---

> > ### Comment · Reviewer_4YKU · 2025-04-02
> >
> > I have read the author's response and find the new experiments broadly compelling. I would prefer to also see overall performance metrics for activation subtraction as well, but this is a lower priority result. That said, the additional experiments have largely mitigated my concerns regarding experiment exhaustiveness. I have raised my score accordingly.

---

> > > ### Author Response · Authors · 2025-04-08
> > >
> > > We want to again thank the reviewer for their time and feedback. The changes regarding over-refusal and overall performance improved the significance of our results greatly. We will further provide the overall performance metrics for activation subtraction for the camera-ready version of our manuscript.

---

### Official Review · Reviewer_5cSG · 2025-03-13

**Overall Recommendation:** 3

**Summary:**

This paper investigates the mechanisms behind refusal behaviors in large language models (LLMs) and discovers that refusals are controlled by multiple- dimension in the model’s activation space. The authors introduce RDO to enhance the refusal capability of large language models (LLMs) through training a learnable refusal subtraction feature. The experiments somewhat illustrate the effectiveness of proposed methods.

**Claims And Evidence:**

While the method improves refusal effectiveness, it also impacts performance on safe instructions. The results in Figure 2 (ASR) and Table 1 (utility) suggest a trade-off between safety and general utility. To better illustrate the advantages and limitations of different methods, it would be beneficial to include an overall performance metric that integrates both ASR and utility.

**Essential References Not Discussed:**

1.	I think over-refusal is an important point for this task, due to the trade-off relationship between the safety and utility. Namely, when enhancing a model’s refusal capability against unsafe samples, over-refusal inevitably emerges on safe samples. There has been a body of literature [1, 2, 3, 4, 5] about over-refusal, discuss and provide some insights for defending over-refusal in this work is necessary.
2.	There is also some work [6] about sparse autoencoders in safety, but it could be regarded as a concurrent work, so not necessary.

**Refs:**

[1] Paul Röttger, et al, “XSTEST: A Test Suite for Identifying Exaggerated Safety Behaviours in Large Language Models”, NAACL, 2024.

[2] Chenyu Shi, et al, “Navigating the OverKill in Large Language Models”, ACL, 2024.

[3] Justin Cui, et al, “Or-bench: An over-refusal benchmark for large language models”, arXiv, 2024.

[4] Hao Li, et al, “InjecGuard: Benchmarking and Mitigating Over-defense in Prompt Injection Guardrail Models”, arXiv, 2024.

[5] Bang An, et al, “Automatic Pseudo-Harmful Prompt Generation for Evaluating False Refusals in Large Language Models”, arXiv, 2024.

[6] Kyle O'Brien et al, “Steering Language Model Refusal with Sparse Autoencoders”, arXiv, 2024.

**Experimental Designs Or Analyses:**

Figure 7 shows that multiple independent directions exist for safe refusal. However, a key question seems undiscovered: what happens if these independent directions are combined as subtract directions? Would this improve or hinder performance? Investigating this could provide further insights into the structure and impact of these refusal directions.

**Methods And Evaluation Criteria:**

The design of gradient-based optimization for discovering refusal directions is overall reasonable and effective.

**Other Comments Or Suggestions:**

1. Minor typos exist, such as in the caption of Figure 6, where there should be a space before “on the right.”

Based on previous issues and concerns, I would like to give weak reject first, but if authors could clarify these concerns, I am willing to raise my rating.

**Other Strengths And Weaknesses:**

Strengths:
1. The paper provide an interesting insight of refusal mechanism in large language model.

Weakness：
1. Figure 2 lacks baseline results, making it difficult to compare performance improvements..
2. The paper does not include an ablation study for the three loss components, which would provide insights into their individual contributions.
3. Although the paper suggests that refusal mechanisms are governed by multiple directions, it does not offer deeper applications or guidance on how this knowledge can be leveraged in LLM safety or adversarial robustness.

**Questions For Authors:**

Please refer to prior comments.

**Relation To Broader Scientific Literature:**

The paper contributes to a broader understanding of LLM refusal mechanisms. It provides evidence that refusal is not governed by a single direction, as suggested in prior work, but rather by a combination of multiple directions. This insight expands the existing literature on LLM safety mechanisms and adversarial attacks.

**Theoretical Claims:**

The claim regarding the independence of refusal directions lacks formal theoretical validation. Further theoretical validation could strengthen the claims. But it is understandable that proving representational independence is a challenging problem due to the modeling difficulty of neural network.

---

> ### Author Rebuttal · Authors · 2025-04-01
>
> We thank the reviewer for the time and effort spent reviewing our manuscript!
>
>
> ### Ablation studies & Safety and Utility Tradeoff
> We added a detailed ablation study, investigating the three loss components. We show the results in [Figures 18 & 19](https://figshare.com/s/4ab2ec422f6bd0262b30) for Llama-3-8B-Instruct. In Fig. 18, we measure how the ASR evolves as we shift the balance between addition and ablation loss weights, requiring $\lambda_{\text{abl}} + \lambda_{\text{add}} = 1$, and observe that performance is quite robust for balanced values (0.2 to 0.8), with both losses being necessary for good generalized performance.
>
> In Figure 19, we ablate the retain loss weight. We use our default ablation and addition weights with different retain weights to train directions, and plot the ASRs compared to the difference in average benchmark performance on MMLU, ARC, GSM8K, and TruthfulQA (here we use the smaller Qwen2.5-3B-Instruct for computational reasons). The ideal direction would have the highest ASR possible for this model, while having no effect on benchmarks or positive effect only so far as preventing the model from refusing benchmark questions. We observe that many combinations of loss weights pareto-dominate the DIM direction for this model, which shows that our approach robustly outperforms the current SOTA.
>
> Further, this figure is in our view the best option to investigate the tradeoff between effectiveness for model manipulation and utility. A combined score will always omit important details. We propose to highlight this figure in the paper to investigate this tradeoff if the reviewer agrees. Here we can see that our approach strictly improves this tradeoff compared to DIM as we pareto-dominate it for both metrics. However, if the reviewer would still like to see a combined score, we will provide one in a new response.
>
> ### Over-Refusal
> We added over-refusal experiments to our manuscript. Due to space constraints, we kindly ask the reviewer to see the rebuttal answer to reviewer 4YKU. We further added a paragraph to the related work section giving the reader an overview of the topic based on the provided references.
>
> ### (Formal) theoretical validation
> We agree that formal validation of the independence would be great. We employ an argument that proves the true causal independence with respect to cosine similarity until the final read-out of the model. We can then use the empirical arguments of related work that directionality to a concept primarily drives the model prediction. For a more in-depth answer please refer to the answer to reviewer NvcZ.
>
> ### Applications
> We agree with the reviewer that outlining practical uses of refusal directions is valuable. We added one paragraph to the manuscript and provided a brief overview here:
> - Most importantly, we aim to emphasize that top-down interpretability not only advances our understanding of AI safety but also deepens our general comprehension of how LLMs represent concepts. Numerous studies have explored the latter, suggesting that any method that enhances this approach could be highly impactful.
> - Offense Applications: Refusal directions can inform (white-box) jailbreak methods, for example, Huang et al. (2024)  in *Stronger Universal and Transferable Attacks by Suppressing Refusals* show that aiming at suppressing DIM refusal directions with input attacks increases their universality and transferability.
> - Defensive Applications: Refusal directions can be used to improve safety via adversarial training (Yu et al., 2024) or inference-time monitoring.
>
>
> ### Other
> **Added No-Intervention to Figure 2:**
> We show the result in [Figures 21-23](https://figshare.com/s/4ab2ec422f6bd0262b30) and we observe that without intervention, the model rarely responds to harmful instructions.
>
> **Combination of Refusal Directions:**
> We thank the reviewer for the great idea and show the result in [Figure 24](https://figshare.com/s/4ab2ec422f6bd0262b30) for the RepInd directions shown in Figure 7. We observe that the composition improves the ASR and using three RepInd directions outperforms DIM. We add this to the main body and believe that this experiment significantly strengthens our manuscript.
>
> **Related Work:**
> We added a paragraph to over-refusal starting on the provided source papers and further list [6] as related work that uses SAEs and thank the reviewer for providing these references.
>
> **Typos:**
> We iterate over the manuscript to remove typos
>
>
> ### Conclusion
> We again want to thank the reviewer for the insightful comments and questions. We believe that addressing the reviewer's concerns greatly improves our manuscript and hope the reviewer agrees with us.

---

> > ### Comment · Reviewer_5cSG · 2025-04-03
> >
> > I appreciate the detailed rebuttals from authors, after carefully reading the authors' response, most of my concerns have been addressed, especially on over-refusal. Although I still have some confusion about theoretical validation of refusal direction independence, I think this paper has provided valuable insights to the community. I am happy to raise my rating.

---

> > > ### Author Response · Authors · 2025-04-08
> > >
> > > We want to again express our gratitude towards the reviewer for this helpful review process. We will make this very clear in the camera-ready version of our manuscript and thank the reviewer for the questions and feedback that helped us to do that.

---

### Decision · Program_Chairs · 2025-05-01

**Decision:**

Accept (poster)

**Comment:**

This paper explores how using adversarial inputs can bypass the internal safety filters of large language models (LLMs). Concretely, the paper explores refusal behaviors in LLMs, finding multiple independent directions and multi-dimensional concept cones that control refusals. It proposes Refusal Direction Optimization (RDO) to identify optimal refusal directions with minimal unintended effects. Experiments show that exploiting these directions enhances attack success rates, indicating complex spatial structures in refusal mechanisms with each direction carrying complementary information. The reviewers unanimously agree that this brings much needed insights into the problem and they support the empirical evidence is sufficient.